

# The role of low-level clouds in the West African monsoon system

Anke Kniffka[1], Peter Knippertz[1], and Andreas H. Fink[1]

[1]Institute of Meteorology and Climate Research, Karlsruhe Institute of Technology, Karlsruhe, Germany

*Correspondence to:* Anke Kniffka (anke.kniffka@kit.edu)

**Abstract.** Realistically simulating the West African monsoon system still poses a substantial challenge to state-of-the-art weather and climate models. One particular issue is the representation of the extensive and persistent low-level clouds over southern West Africa (SWA) during boreal summer. These clouds are important in regulating the amount of solar radiation reaching the surface but their role in the local energy balance and the overall monsoon system has never been assessed. Based
on sensitivity experiments using the ICON model for July 2006, we show for the first time that rainfall over SWA depends logarithmically on the optical thickness of low clouds, as these control the diurnal evolution of the planetary boundary layer, vertical stability and finally convection. In our experiments, the increased precipitation over SWA has small direct effects on the downstream Sahel, as higher temperatures due to increased surface radiation are accompanied by decreases in low-level moisture due to changes in advection, leading to almost unchanged equivalent-potential temperatures in the Sahel. A systematic
comparison of simulations with and without convective parameterisation reveals agreement in the direction of the precipitation signal but larger sensitivity for explicit convection. For parametrized convection the main rainband is too far south and the diurnal cycle shows signs of unrealistic vertical mixing, leading to a positive feedback on low clouds. The results demonstrate that relatively minor errors, variations or trends in low-level cloudiness over SWA can have substantial impacts on precipitation. Similarly they suggest that the dimming likely associated with an increase in anthropogenic emissions in the future would lead
to a decrease of summer rainfall in the densely populated Guinea Coastal area. Future work should investigate longer-term effects of the misrepresentation of low clouds in climate models, e.g. moderated through effects on rainfall, soil moisture and evaporation.

*Copyright statement.* TEXT

## 1 Introduction

Modelling the West African monsoon (WAM) system is a challenge (Roehrig et al., 2013). Climate and weather models show a considerable inter-model spread when studying for example the influence of sea surface temperatures ($SST$s) on the WAM circulation (Xue et al., 2010; 2016; Rodríguez-Fonseca at al., 2015), interactions with the land surface (Boone et al., 2009) or the representation of the hydrological cycle (Meynadier et al., 2010; Poan et al., 2016). Current numerical weather prediction (NWP) models do not show skillful forecasts of precipitation for the next days (Haiden et al., 2012, Vogel et al., 2018).



The climate of West Africa is to a large extent controlled by the WAM (Hall and Perrilé, 2006; Fink et al., 2017). The monsoon is connected to the large north–south pressure gradient between higher pressure over the Atlantic cold tongue (Caniaux et al., 2010), which develops during March to May, and the Saharan heat low forming due to the enhanced insolation in northern hemispheric summer. The onset of the monsoon in June (Fitzpatrick et al., 2015), which often occurs abruptly (Sultan and Janicot, 2000), is accompanied by an increase in southwesterly inflow from the tropical Atlantic and a northward shift of the main rain band and the Intertropical Discontinuity (ITD), the airmass boundary between cool monsoonal and hot and dry Saharan air. The rain band reaches its maximal northern position in August/September, after which the rainband and ITD shift southward again. Due to this characteristic seasonal behaviour, local variations in rainfall, winds, temperature and clouds are determined by the WAM system (Thorncroft et al., 2011). Eltahir and Gong (1996) developed a theoretical framework for the driving forces of the WAM describing it as a direct thermal circulation for moist atmospheres. They found that the gradient of entropy in the planetary boundary layer (PBL) is a key factor for the strength of the monsoon circulation and its inter-annual variations. Using a simple 2D-model, Zheng et al. (1999) argue that an increase of net surface radiation leads to an increased entropy and thus a stronger WAM circulation. Several studies stress the importance of low-level processes for the development of the WAM (Perillé et al., 2016; Eltahier and Gong, 1996).

Variability within the WAM and day-to-day changes are determined by more local factors such as surface characteristics and incoming solar radiation (Lafore et al., 2017; Taylor et al., 2011) or specific regional features such as orography or the land-sea breeze. Lavender et al. (2010) studied soil-moisture and land-atmosphere coupling for the 15-day westward-propagating mode of intraseasonal variability of precipitation and wind, and found that soil moisture plays an active role in the development of the WAM system. Propagating synoptic-scale disturbances such as African easterly waves or single vortices can lead to marked variations in rainfall (Diedhiou et al., 1999; Knippertz et al., 2017). A key process for many aspects of the WAM is moist convection, which occurs in a wide range of degrees of organization depending on ambient thermodynamic conditions and shear (Maranan et al., 2018). Marsham et al. (2013) demonstrated that the use of a convective parameterization can lead to substantial errors in the diurnal cycle of precipitation, cloudiness and the entire monsoon circulation due to differences in both latent and cloud radiative heating. Couvreux et al. (2014) assessed the diurnal cycle of thermodynamics in the lower troposphere in four contrasted regimes over West Africa. The NWP models they analyze suffer from an erroneous surface-atmosphere-cloud coupling on short time scales, leading to false cloud cover, particularly in the lower parts of the atmosphere. Not limited to West Africa, Noda et al. (2009) show that sub-grid cloud processes in the Non-Hydrostatic Icosahedral Atmospheric Model (NICAM) influence not only the development of low-level cloudiness but also middle and higher clouds, even at horizontal grid scales of 14 and 7 km due to differences in turbulent transport. Also different radiation schemes have been found to impact on precipitation and the north-south gradient of surface temperature, which affects the strength of the monsoon flow (Li et al., 2015).

An interesting local to regional-scale feature, which is currently gaining increasing attention, is the low-level stratiform cloud cover in southern West Africa that develops at night-time and persists long into the following day (Knippertz et al., 2011; Schrage and Fink, 2012; Schuster et al., 2013). Due to this persistence, the radiative characteristics of these clouds influence the PBL development at the Guinea Coast and further inland. Its formation is connected to the evolution of the nocturnal



low-level jet (NLLJ; Schrage et al., 2007) and involves advection of cool air from the ocean, radiative cooling and turbulent mixing associated with the NLLJ (Schuster et al., 2013; Adler et al., 2017). During the monsoon season low-level stratus occurs frequently with typically less than a third of all nights being cloud-free at a given location (Schrage and Fink, 2012; van der Linden et al., 2015; Kalthoff et al., 2018). Climate models struggle to realistically represent this phenomenon correctly in

terms of cloud amount and occurrence as well as wind speed (Knippertz et al., 2011; Hannak et al., 2017). Hill et al. (2018) studied the radiative impact of different cloud types in this region with detailed radiative transfer calculations based on the CERES-CloudSat-CALIPSO-MODIS dataset (Ham et al., 2017) using the two-stream radiative transfer model SOCRATES (Suite Of Community RAdiative Transfer codes based on Edwards and Slingo; Edwards and Slingo, 1996). They find that low-level clouds have a cooling effect, the magnitude of which depends on the overlying midlevel and high clouds. Ignoring

low-level clouds (defined as below 680 hPa by Hill et al.) but keeping all other clouds the same would lead to errors of about 35 $\mathrm{Wm}^{-2}$ for downwelling surface solar irradiance ($SSI$) and -25 $\mathrm{Wm}^{-2}$ for outgoing shortwave radiation ($OSR$) at the top of the atmosphere (TOA). Knippertz et al. (2011) indeed found that the lack of low-level cloudiness in climate models leads to an overestimation of $SSI$ compared to station measurements but feedbacks were not considered explicitly. It can be expected that increased surface heating due to a lack of low clouds should lead to a deeper PBL and possibly more convection,

which may significantly redistribute moisture vertically. This would be consistent with recent findings by Deetz et al. (2018a), who demonstrate significant sensitivity in PBL height and daytime stratus-to-cumulus transition to aerosol radiative effects. In addition, mis-representing low clouds is likely a source of error in the simulated moisture budget (Schrage and Fink, 2012), which together with $SST$s controls the WAM development to a large extent (Xue et al., 2010; 2016).

To the best of our knowledge, this study is the first to analyze the radiative impact of the low-level cloudiness over southern

West Africa on the thermodynamics and dynamics of the regional atmospheric system in a fully non-linear and systematic way. The analysis is based on a number of targeted sensitivity experiments using the numerical weather prediction model ICON (Icosahedral Nonhydrostatic), systematically changing the optical thickness of the model clouds. This allows us to clarify the impact of the inter-model spread in cloudiness found in Hannak et al. (2017) on the overall monsoon development. Effects of convective parametrization on the sensitivity will be tested. This study is part of the Dynamics-Aerosol-Chemistry-Cloud

Interactions in West Africa (DACCIWA) project (Knippertz et al., 2015) that aims to better understand the consequences of the rapid increase of anthropogenic emissions in West Africa on the local air quality, weather and climate. Although aerosols are not directly modeled in our experiments, the effects found for imposed changes of cloud optical thickness also help to understand variations in the natural system brought about by aerosol effects on cloud properties and radiation, which in a similar way control the amount of shortwave radiation reaching the surface or interact with clouds through modifications in the

diurnal cycle of the PBL (e.g. Deetz et al., 2018a).

This article is structured as follows: in Sect. 2 the data and methods are introduced together with a description of the ICON model and the experimental design. The results of the sensitivity experiments are presented in Sect. 3, where we first consider the thermodynamic and dynamic effects on the southern West African region, where we modify clouds, and later expand the analysis to the greater WAM region including the Sahel. The results are further discussed and summarized in the concluding

Sect. 4.



## 2    Data and methods

This section first details the observational data (ground- and space-based) used as a reference for our modeling experiments (Sect. 2.1) followed by a general description of the ICON model and the design of the sensitivity experiments (Sect. 2.2). The analysis will concentrate on July 2006 and spatially on the DACCIWA study region ($5° - 10°N, 8°W - 8°E$, visualized in Fig.
1), as used in several related papers (e.g. Hannak et al., 2017; Hill et al., 2018). July 2006 was characterised by a relatively late monsoon onset as documented for example in Janicot et al. (2008).

### 2.1    Observational data

#### 2.1.1    Precipitation

Precipitation information from two different sources are considered in this study. The first is the Tropical Rainfall Measuring
Mission (TRMM) 3B42 version 7 dataset. TRMM is a joint mission of the National Aeronautics and Space Administration and the Japan Aerospace Exploration Agency covering the tropical and subtropical regions of the earth during 1997–2015. This dataset is created with the TRMM Multisatellite Precipitation Analysis method (Huffman et al., 2007) combining the TRMM precipitation radar with measurements from microwave and infrared sensors on several low earth orbiting and geostationary satellites, and is calibrated with rain gauge data on a monthly basis. The rainfall data used in this study were aggregated from
3-hourly measurements on a $0.25° \times 0.25°$ grid.

In addition to TRMM, rainfall from the Global Precipitation Climatology Project (GPCP) was used. GPCP combines several sources of rainfall measurements into one global dataset with a high data density and accuracy. It was established by the World Climate Research Programme to quantify the distribution of precipitation around the globe on climatological time scales (Adler, 2003). In GPCP, ground-based rain gauge measurements as well as satellite-based precipitation estimates are combined to give
a merged product. The rain gauge measurements stem from the Global Precipitation Climatology Centre monitoring product of the German Weather Service (DWD). The satellite data consist of infrared and microwave radiance-derived rainfall estimates from geostationary as well as polar orbiting satellites. We used daily data in $1.0° \times 1.0°$ horizontal resolution.

#### 2.1.2    Radiation

$SSI$ measurements stem from the climate data record SARAH (Surface Solar Radiation Data Set Heliosat) version 2. It
was created by the Satellite Application Facility on Climate Monitoring (CM SAF) based on Meteosat Visible and Infrared Imager (MVIRI) and Spinning Enhanced Visible and InfraRed Imager (SEVIRI) measurements on the geostationary Meteosat satellites (Müller et al., 2015). From MVIRI, the broadband visible channel and from SEVIRI the channels 0.6 and 0.8 $\mu$m are used. SARAH was produced using a retrieval system based on the Heliosat method and an efficient clear-sky surface solar radiation transfer model (Mueller et al., 2009; Posselt et al., 2012). For this study we use the monthly mean products of the
dataset with a horizontal resolution of $0.05° \times 0.05°$. In addition, we employ the much-coarser EBAF-Surface Ed4.0 dataset (Energy Balanced And Filled) containing monthly averaged $SSI$ fields with a horizontal resolution of $1° \times 1°$. This product is



based on the CERES (Clouds and the Earth's Radiant Energy System) algorithm (Loeb et al., 2009; Young et al., 1998), which uses information from the CERES shortwave broadband radiometers but also from instruments on geostationary satellites to account for the diurnal variability in the data. Several CERES instruments are mounted on polar orbiting satellites such as TRMM, Terra, Aqua and NPP (Suomi National Polar-orbiting Partnership). To derive the radiative fluxes at the surface, cloud

imager data for scene classification, cloud physical properties, temperature, water vapor, ozone and aerosol data as well as a broadband radiative transfer model are needed.

The satellite-derived $SSI$ fields are complemented with a small set of surface measurements. Unfortunately, there are very few ground-based measurements of $SSI$ available in the DACCIWA study region during July 2006. South of $10°N$, only the stations Lamto (Ivory Coast, $6.22°N, 5.03°W$), Cotonou and Parakou (both Benin, at $6.35°N, 2.43°E$ and $9.33°N, 2.62°E$,

respectively) delivered gap-free measurements from standard instruments, i.e. a Gunn-Belani radiometer (Lamto) and CNR1 radiometers from Kipp & Zonen (Parakou and Cotonou).

For $OSR$ at TOA, monthly mean averages from the dataset GERB/SEVIRI ed. 2.0 from CM SAF (Clerbaux et al., 2017) were used. GERB is the geostationary earth radiation budget instrument onboard Meteosat Second Generation satellites (Harries et al., 2005). This broadband radiometer is designed to measure the earth's total emitted longwave and solar reflected

radiances with high temporal resolution (5 min) and 50 km grid-spacing. It is available as TOA reflected shortwave and TOA emitted thermal fluxes. In the present study, we consider only the shortwave flux. For this dataset, SEVIRI measurements are employed to refine GERB's original spatial resolution to a $0.1° \times 0.1°$ grid. In addition, the $1° \times 1°$ monthly EBAF-TOA Ed4.0 dataset for shortwave radiation is used that has been derived using the same CERES algorithm as for the surface.

## 2.2 Modeling experiments

### 2.2.1 General model description

The highly scalable ICON model (Zängl et al., 2014) was recently developed by the Max Planck Institute for Meteorology and the DWD, and became DWD's new operational global NWP in January 2015. ICON's horizontal Arakawa C type grid is based on triangles, which cover the globe with approximately equal area everywhere, and allows easy nesting. The vertical coordinate is height-based and terrain following in the lower levels but smoothed in the upper troposphere via the application of a SLEVE

(smooth level vertical) coordinate (Leuenberger et al., 2010). For the dynamical core the continuity equation is formulated in the flux form with density as the prognostic variable, enabling exact local mass conservation. The equations are solved non-hydrostatically and the time integration is performed with a two-time-level predictor-corrector scheme. Apart from the sound wave propagation, this scheme is fully explicit. The fast physics packages are inherited from the Consortium for Small-scale Modelling (COSMO) model (Doms and Schättler, 2004) but are partly reformulated for ICON. The cloud microphysics scheme

is the COSMO-EU five-category prognostic scheme (Doms and Schättler, 2004; Seifert, 2008) with the extension of ice sedimentation. The turbulence scheme by Raschendorfer (2001) solves the prognostic equation for turbulent kinetic energy ($TKE$) and for the land-surface interaction TERRA (Heise, 2006) is used in an updated version. The slow physics parametrizations correspond to those from the Integrated Forecasting System (IFS) of the European Centre for Medium-Range Weather Fore-





casts (ECMWF): the Bechtold et al. (2008) convection scheme, the Lott and Miller (1997) subgrid-scale orography scheme and the Orr et al. (2010) non-orographic gravity-wave drag scheme. Radiative transfer is solved with the Rapid Radiation Transfer Model (RRTM, Mlawer et al., 1997), where a greens function approach is applied for solar bands with approximated diffuse radiation (Barker et al., 2002).

All simulations in this paper were initialized with ERA Interim data (ERA-I hereafter), ECMWF's global atmospheric reanalysis (Dee et al., 2011) and do not use data assimilation. ERA-I is created by assimilating all available measurements into a single forecast model environment, resulting in a multivariate, spatially complete and coherent record of the global atmospheric state. ERA-I data are used in the highest possible horizontal resolution of about 80 km and with 60 vertical levels up to 0.1 hPa. Typically ERA-I contains most observations at 12 and 00 UTC. Initializing ICON runs at 00 UTC would mean
to start directly in the development phase of the low-level clouds and therefore 12 UTC was preferred as an initialization time.

### 2.2.2   Design of experiments

To assess the impact of variations in cloudiness in the ICON model, a series of experiments was designed. In these, the original cloud liquid water content $q_c$ in the DACCIWA study region and below 700 hPa is manipulated immediately before the call of the radiation scheme by multiplying it with an opacity factor $f_{op}$ to mimic an increase or decrease of the low clouds' optical
thickness. After that, $q_c$ is set back to the original value and the model is allowed to run freely until the next call of the radiation routine. In this way it is ensured that only the radiation can impact on the dynamics and thermodynamics, creating changes in temperature $T$, relative humidity $RH$ and winds etc., which in turn can influence the development of clouds itself. $f_{op}$ is varied from 0.1 to 10. The low values are at the extreme end of cloud underrepresentation found in Hannak et al. (2017), while $f_{op} = 10$ should be regarded as a somewhat unrealistic sensitivity test.
Two sets of experiments were performed with ICON:

1. PARAM: For this set ICON was run in the current operational global setting with a grid spacing of 13.2 km grid and 91 vertical levels. Integration time is five days. $f_{op}$ is varied in eight steps from 0.1 to 10.0 to systematically analyze the effect of low-level clouds. Due to the relatively high computational costs, runs are restricted to July 2006 and only started every $4^{th}$ day in order to have one day of overlap between the simulations. All in all $8 \times 8$ 5-day simulations
25       were performed for this set.

2. EXPL: The overall setting is identical to PARAM, but another nest was added to achieve 6.6 km horizontal resolution, which allowed switching off the convection scheme. In order to keep the amount of data manageable, only two $f_{op}$ values were run: 0.1 and 1.0. This will show whether the sensitivities found for PARAM depend on the convection scheme, as demonstrated for example for the larger WAM circulation by Marsham et al. (2013). One may argue that 6.6 km is still
30       too coarse for explicit convection, but Marsham et al. (2013) showed that for West Africa explicit convection even at a grid-spacing of 12 km improves the diurnal cycle of the PBL and convection.



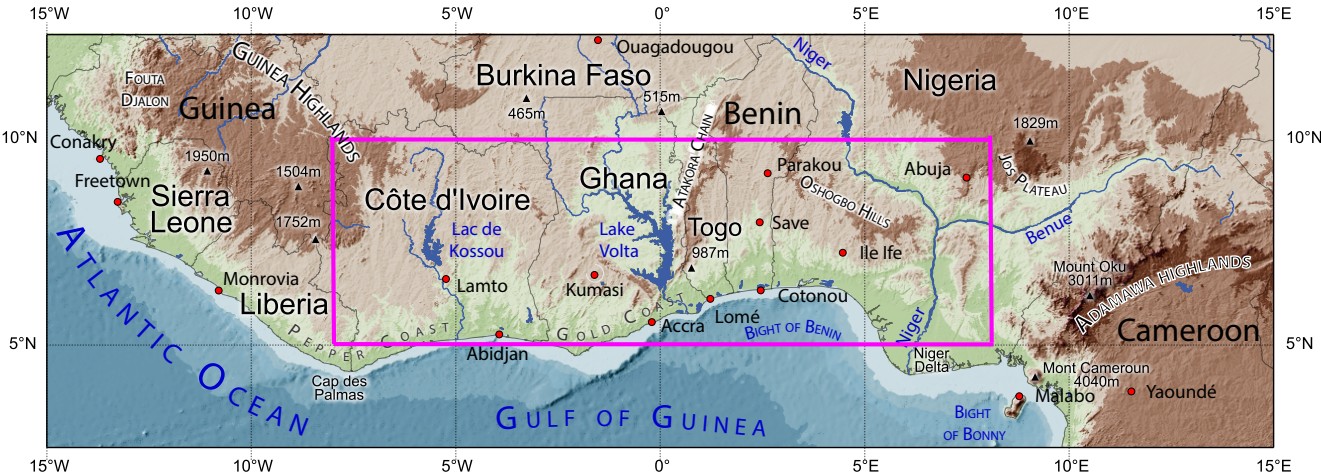

**Figure 1.** Map of southern West Africa indicating the geographical locations referenced in the text. Low-level clouds were modified within the pink square.

## 3 Results

In this section we will discuss the outcome of the control and sensitivity experiments. The analysis will be broken down in four parts. The first (Sect. 3.1) will concentrate on a general model evaluation over West Africa comparing ICON PARAM and EXPL with observations. Sect. 3.2 analyses diurnal-mean responses over the DACCIWA study region considering the full range of $f_{op}$. Sect. 3.3 discusses the impact of cloud modification on the diurnal cycle covering a wide range of parameters including precipitation, clouds, temperature and humidity for southern West Africa, while the following Sect. 3.4 will analyze impacts on the wider WAM region. Sects. 3.2–3.4 also contain a systematic comparison between the PARAM and EXPL experiments. A geographical map of southern West Africa indicating the study region and locations mentioned in the analysis is shown in Fig. 1.

## 3.1 Model evaluation

Here a characterization of the meteorological conditions in southern West Africa for the wet monsoon month July 2006 is given concentrating on precipitation and radiation. Comparison of ICON runs with observations will reveal the applicability of the ICON model for the following experiments and the sensitivity to convective parametrization.

Figure 2 shows July 2006 averaged daily precipitation for ICON EXPL, ICON PARAM, TRMM and GPCP together with the respective averages over the DACCIWA region as numbers. TRMM and GPCP are shown in their native resolutions, while ICON EXPL and ICON PARAM are interpolated to grids with $0.0625° × 0.0625°$ and $0.125° × 0.125°$ spacings, respectively. All four datasets have marked local maxima over the Niger Delta region in Nigeria and adjacent Adamawa Highlands as well as along the coast of Guinea, Sierra Leone and Liberia and adjacent Guinea Highlands. Within our main region of interest, there



**Figure 2.** Mean daily rainfall for July 2006 over the larger West African domain for (a) ICON EXPL, (b) ICON PARAM as well as the observational datasets (c) TRMM and (d) GPCP with averages over the DACCIWA box (marked with green lines) on top of each panel.

are substantial differences with respect to the position of the main rainband. The two observational datasets, TRMM and GPCP, consistently show a well defined zonal rainband stretching across the Sahel with substantially drier conditions over southern West Africa and the adjacent Atlantic Ocean (Figs. 2c and d). There is, however, some conspicuous disagreement between the two in coastal areas, where satellite retrievals are complicated by the sharp change in surface properties, illustrating the overall observational uncertainty, which is also related to the (relatively sparse) ground-based network. ICON EXPL produces a much wetter, northward shifted main rainband compared to ICON PARAM with a lot of fine structure related to the high spatial resolution (Fig. 2a). In stark contrast, ICON PARAM struggles to represent the shift of rainfall inland resulting in substantially lower amounts in the Sahel (Fig. 2b). Within the DACCIWA box area-averaged rainfall agrees within less than 10% between the observational datasets. Despite the overall dry bias of ICON PARAM, agreement with observations in the DACCIWA box is





satisfactory, while ICON EXPL underestimates rainfall by on the order of 30% (3.3 mm day$^{-1}$ vs. 4.7 mm day$^{-1}$ for TRMM and GPCP combined). At least some of the patterns within the DACCIWA box (e.g. slightly moister northwestern corner over Ivory Coast, drier Lake Volta region and a local maximum over the Atakora chain) are consistent between all four datasets. This comparison reveals an enormous sensitivity of the WAM to convective parametrization. In agreement with Marsham et al.

(2013) explicit convection creates substantially more rainfall but the northward shift we observe for ICON was not found for the Unified Model used in that study. Ultimately, the low agreement between the two ICON simulations and with observations hampers drawing rigorous quantitative conclusions from our sensitivity experiments and forces us to analyse all subsequent aspects separately for PARAM and EXPL. However, the errors in latitudinal position and intensity of the Sahelian rainband we find here are commonplace in intercomparison studies for climate models (Mohino et al., 2011; Roehrig et al., 2013) and

allow interfering whether the sensitivities we find are robust against these differing model basic states.

Similar to Fig. 2, Fig. 3 shows comparisons between ICON EXPL and PARAM with the observational datasets CM SAF and CERES for $SSI$ (left) and $OSR$ (right), again in their native resolution with DACCIWA-box averages provided as numbers. Additionally, surface radiation measurements from the ground stations in Lamto, Cotonou and Parakou are included for comparison. The depiction is limited here to the DACCIWA box, as this is where our main interest in clouds lies. $SSI$ depends

on how much sunlight is absorbed or reflected on its way through the atmosphere, mostly by clouds but also by aerosols. This is clearly illustrated in the high-resolution datasets, ICON and CM SAF, where the relatively cloud-free western Bight of Benin and Lake Volta area show local maxima (Figs. 3a–c). All datasets reveal a general tendency for lowest $SSI$ in the inland "stratus belt" around 7°N and an increase towards the less cloudy Sahel in the north. Minima are usually found over southwestern Nigeria with values dropping to below 120 W m$^{-2}$. In addition to many smaller differences in pattern, there are

quite considerable deviations in absolute values between the four datasets.

ICON EXPL shows the lowest $SSI$ values with an area average of 164.7 W m$^{-2}$ (Fig. 3a), much lower than PARAM with 191.6 W m$^{-2}$ (Fig. 3b). We will see later in this paper that there likely is a direct connection between this and the much lower rainfall found in EXPL through an increase in vertical stability due to less sunlight reaching the ground. Evaluating this with observations is a challenge due to the many assumptions made in satellite-derived $SSI$ and the few surface observations.

CM SAF shows an overall similar pattern as the two ICON simulations but with systematically higher values inland and an area average of 204.3 W m$^{-2}$ (Fig. 3c). This is clearly at odds with the ground stations and is likely due to the method of determining the range of minimum and maximum irradiance for the applied self-calibration. The surface albedo should correspond to the lowest irradiance measurement found per pixel in a given time period, since clouds appear brighter than the surface (except for snow) but in this region is likely still contaminated by clouds. Therefore it suggests an unrealistically

bright surface (see also discussion of this problem in Hannak et al., 2017). In contrast, CERES does not seem to suffer from this problem due to a different retrieval strategy (Fig. 3d). The box-averaged $SSI$ is 188.4 W m$^{-2}$ and therefore very close to the ICON PARAM value, although with much less fine structure. Overall this analysis demonstrates a significant observational uncertainty and suggests an overestimation of clouds in ICON EXPL, while ICON PARAM fields look more consistent with observations.



**Figure 3.** Mean July 2006 $SSI$ over the DACCIWA box from (a) ICON EXPL and (b) ICON PARAM as well as the satellite-derived datasets (c) CM SAF and (d) CERES plus station data as filled circles. Corresponding $OSR$ fields are given in (e)–(h). Area averages are provided on top of each panel.

The right panels in Fig. 3 show corresponding fields of $OSR$. Given that this quantity can be measured directly from satellite, it is no surprise that the agreement between the two observational datasets is much closer, apart from, of course, the obvious



**Figure 4.** Averages over July 2006 and the DACCIWA box of $SSI$ (a), $T$ at 950 hPa (b), precipitation ($RR$) (c), $SLI$ (d), $OSR$ (e) and $OLR$ (f) depending on the opacity factor $f_{op}$ plotted with an exponential scale. ICON PARAM is depicted with solid blue lines, while the dashed cyan lines denote ICON EXPL (see Sect. 2.2.2). The thin grey line marks the position of the control run $f_{op} = 1.0$.

differences in resolution (Figs. 3g and h). Nevertheless, even here there is a non-negligible observational uncertainty with the area averages differing by 3.3 W m$^{-2}$, corresponding to 2%. There are many structural similarities to $SSI$ (left panels of Fig. 3) but with the opposite sign, indicating that clouds suppress $SSI$ but increase $OSR$ due to their high reflectivity. Consistently, ICON EXPL shows the highest area-averaged $OSR$ of 153.1 W m$^{-2}$ (Fig. 3e). In contrast, ICON PARAM produces much lower values of only 130.6 W m$^{-2}$ (Fig. 3f). Given an $SSI$ similar to CERES, this suggests an overestimation of scattering on cloud droplets, i.e. biases in the amounts of cloud water or ice or their size distributions. This comparison reveals that the substantial differences between PARAM and EXPL found for precipitation also hold for cloud radiative effects and that the dissatisfying agreement with observations somewhat limits the quantitative interpretation of our sensitivity experiments.





## 3.2 Dependence of diurnal mean fields on $f_{op}$

In this section, first results for the modifications of $f_{op}$ in ICON (see Sect. 2.2.2) will be presented for PARAM and EXPL. Parameters considered for this investigation are precipitation, $SSI$ and $OSR$ as in Sect. 3.1 and additionally temperature at 950 hPa $T_{950}$, outgoing longwave radiation ($OLR$) and surface longwave irradiance ($SLI$), all averaged over the DACCIWA

box as in Fig. 3. The questions to be addressed in this section are: (a) How is the sensitivity of the considered parameters to $f_{op}$? (b) How do the fully nonlinear signals found in ICON differ from the purely radiative transfer computations by Hill et al. (2018)? (c) To what extent does the signal depend on the use of a convective parametrization (comparing PARAM with EXPL)?

In PARAM, $SSI$ decreases largely logarithmically with increasing optical thickness (Fig. 4a) ranging from 158.2 W m$^{-2}$

to 236.9 W m$^{-2}$. Only at the highest $f_{op}$ of 10 is there a clear indication for a certain "saturation" of the signal. Given this behavior in $SSI$, it is to be expected that $T_{950}$ also decreases with $f_{op}$ (Fig. 4b). The small range, however, of less than 0.5°C (23.5–24.0°C) suggests that some of the additional radiative heating of the surface is balanced by transports into the atmosphere, i.e. either a deeper PBL or convection. This is consistent with the flatter curve at the lowest $f_{op}$ values. Figure 4c demonstrates that the effects on precipitation are in fact enormous, leading to a doubling in daily precipitation from 3.2 mm

for $f_{op} = 10$ to 6.3 mm for the optically thinnest clouds with $f_{op} = 0.1$. The shape of the curve is very similar to that of $SSI$ (Fig. 4a), indicating a strong control of radiation on convective initiation.

With respect to the other components of the radiative budget, Fig. 4d shows that $SLI$ is hardly affected varying between 412.5 and 409.8 W m$^{-2}$ only, which corresponds to less than 0.7%. This low sensitivity is the result from small variations in low-level temperature (Fig. 4b) and an overall very moist atmosphere that traps longwave radiation , almost irrespective

of low-level clouds. At TOA, both longwave and shortwave outgoing radiation increase with increasing $f_{op}$ (Figs. 4e and f). Again, the variation in shortwave radiation dominates over that in the longwave (from 94.4 to 157.5 W m$^{-2}$ and from 228.2 to 243.6 W m$^{-2}$, respectively). The increase in $OSR$ is consistent with the increased reflection from low-level clouds, as already discussed in the context of Fig. 3. The difference in $SSI$ and $OSR$ signals shows that extinction increases with increasing $f_{op}$. As will be seen later, this extinction is caused by scattering on cloud droplets and absorption of water vapour. The increase in

$OLR$ is consistent with the decrease in precipitation (Fig. 4c) associated with less deep convective clouds.

The recent study by Hill et al. (2018) mentioned in the Introduction allows a rough estimate of how much of the signals found in Fig. 4 is due to direct radiative effects and how much is due to the dynamical response of the system. Ignoring all clouds below 680 hPa, their radiative transfer calculations for June–September 2006–2010 yield the following signals: Increases of 35 W m$^{-2}$ in $SSI$ and of 2 W m$^{-2}$ in $OLR$ as well as decreases of 25 W m$^{-2}$ in $OSR$ and of 11 W m$^{-2}$ in

$SLI$. Comparing these values with differences between $f_{op}$ of 1.0 and 0.1 in Fig. 4 shows that the ICON PARAM-generated responses in shortwave radiation for July 2006 have a larger amplitude. Given the reasonable agreement with CERES in $SSI$ (Fig. 3) and the slightly shallower layer of cloud modification (below 750 hPa vs. below 680 hPa), this is a surprising result. The most plausible explanation is that the relatively dry July 2006 had overall less mid- and high-level clouds than the June–September 2006–2010 average, leading to a relatively larger effect of low-level cloudiness (consistent with Fig. 9 in Hill et al.,





2018). The dynamical response of the atmosphere is more evident in the longwave component in ICON PARAM. The increase in deep convection with optically thinner low clouds leads to a decrease in $OLR$ in the model on the order of 10 W m$^{-2}$, while the radiative transfer calculations by Hill et al. show a small increase. In contrast, the increase in low-level temperature, deep convective clouds and column moisture (see Fig. 12) leads to almost constant $SLI$ in the model, while the purely radiative
effect would be a marked decrease.

Finally, the differences between PARAM and EXPL in Fig. 4 illustrate the sensitivity of the response to horizontal resolution and the use of convective parametrization. The overall behavior of EXPL (dashed lines in Fig. 4, $f_{op}$ values of 1.0 and 0.1 only) is comparable but there are deviations in terms of basic state and sensitivity. As already discussed, EXPL has more clouds, leading to lower $SSI$ and higher $OSR$ (both on the order of about 20 W m$^{-2}$; Figs. 4a and e). Interestingly, the
low-level temperature is almost identical for $f_{op} = 1.0$ but slightly warmer in EXPL for $f_{op} = 0.1$ (Fig. 4b), indicating subtle differences in the surface energy budget. Despite the warmer temperatures, precipitation is always lower than in PARAM (Fig. 4c), suggesting that convection is less easily triggered in EXPL (daily sums are 3.3 and 6.1 mmh$^{-1}$ for $f_{op} = 1.0$ and $f_{op} = 0.1$, respectively). This could be an explanation for the overall higher sensitivity in EXPL, making the simulation even more dependent on modifications of solar radiation reaching the ground. With respect to longwave components (Figs. 4d and
f) EXPL shows higher $SLI$ and higher $OLR$ (about 8 and 20 W m$^{-2}$, respectively). The former is consistent with more low-level clouds for $f_{op} = 1.0$ and warmer low-level temperatures for $f_{op} = 0.1$. The latter mirrors the reduced ice content of EXPL compared to PARAM in the upper levels of the troposphere (see right panels of Fig. 6), which facilitates the escape of longwave radiation to space and therefore enhances $OLR$.

### 3.3 Impact on the diurnal cycle

In this section we will continue analyzing the effect of modifying the optical thickness of low clouds, but here with a focus on the diurnal cycle. The analysis begins with impacts on precipitation and clouds followed by an investigation of the vertical structure of the signal.

### 3.3.1 Precipitation and clouds

For precipitation, PARAM generally shows a distinct maximum at 15 UTC (corresponding to local time in our study region)
and lowest rainfall in the second half of the night (Fig. 5). Consistent with Fig. 4c, a decrease in $f_{op}$ leads to a monotonic and smooth increase in precipitation at all times of day, apart from the early morning hours, when the effect is weak. At the time of maximum precipitation, the rainfall from experiment $f_{op} = 0.1$ is 2.5 times larger than that for $f_{op} = 10.0$. The morning onset of rainfall is earlier for low $f_{op}$, as the buildup of instability due to incoming solar radiation occurs faster after sunrise. EXPL shows some significant differences (blue lines in Fig. 5). The diurnal peak is shifted to 18 UTC, as it takes more time
to trigger convection without a parametrization (Marsham et al., 2013). This corresponds much better to the typical timing of precipitation observed in this area (Kalthoff et al., 2018) and to the TRMM observations included in Fig. 5, despite the overall large bias already discussed (Fig. 2). The onset of precipitation is not strongly affected by $f_{op}$ in EXPL but the cessation is, with convection persisting much longer into the night for the optically thinnest low-level clouds, suggesting a much higher





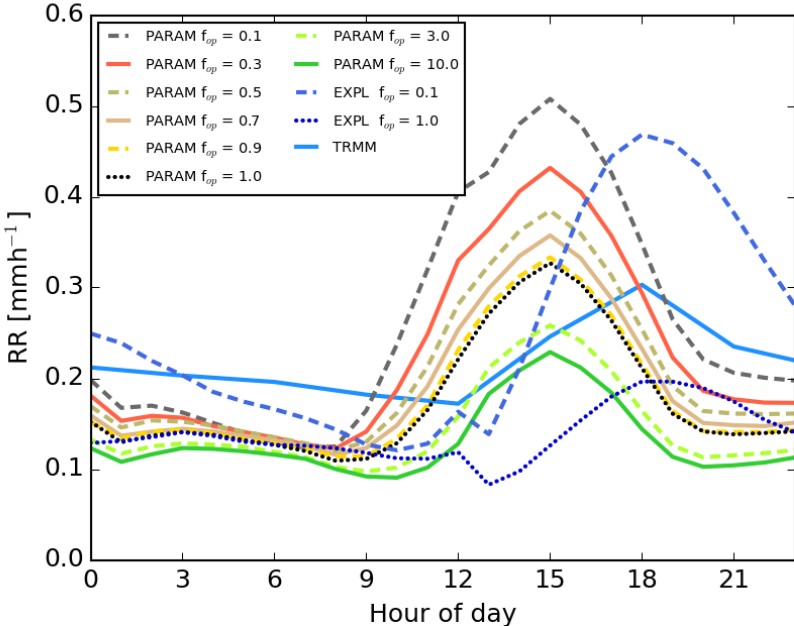

**Figure 5.** Diurnal cycle of precipitation averaged over the DACCIWA box and for July 2006. Different lines show TRMM observations and PARAM and EXPL simulations for varying $f_{op}$.

degree of organization. We have no explanation for the kinks in the curves around 12 UTC in EXPL and therefore attribute those to insufficient sampling. In terms of the diurnal maxima, values for EXPL are systematically lower with 0.17 and 0.47 mm h$^{-1}$ for $f_{op} = 1.0$ and $f_{op} = 0.1$, respectively, compared to 0.32 and 0.5 mm h$^{-1}$ for PARAM.

Figure 6 shows the diurnal cycle in the vertical structure of cloud cover $CLC$, cloud water content $q_c$ and cloud ice content $q_i$

for PARAM and EXPL and for $f_{op}$ values of 0.1 and 1.0. PARAM shows a clear three-layer cloud structure at all times of day as documented for other tropical regions (e.g. Johnson et al., 1999). Low-level clouds are mostly confined to below 750 hPa with a relatively minor mid-level cloud layer around 500–600 hPa. While the former contain significant amounts of $q_c$ (middle column of Fig. 6), the midlevel clouds also contain some cloud ice (right column of Fig. 6). In addition, a substantial high-level cloud cover between 400 and 100 hPa containing significant amounts of cloud ice is simulated in PARAM. In particular the

low and high clouds show a distinct diurnal cycle. At 00 UTC the low-level cloud deck is beginning to form, reaching a sharp peak around 950 hPa at 06 UTC accompanied by a corresponding increase in $q_c$ (Figs. 6a and b). At midday (Fig. 6c), radiative heating lifts and dissolves the low-level cloud deck shifting the maximum in $CLC$ and $q_c$ to 850 hPa (Fig. 6c). Finally by 18 UTC (Fig. 6d) daytime heating and mixing have reduced $CLC$ and $q_c$ to create a diurnal minimum. This general diurnal behavior in low-level cloudiness in PARAM resembles that found in ECMWF analysis data (see Hannak et al., 2017). Midlevel

clouds do not show pronounced diurnal variations but also have a minimum in $CLC$ and $q_c$ at 18 UTC, possibly suggesting similar mechanisms as for the low clouds. High-level $CLC$ and $q_i$ are lowest at 12 UTC and highest at 00 UTC, when they





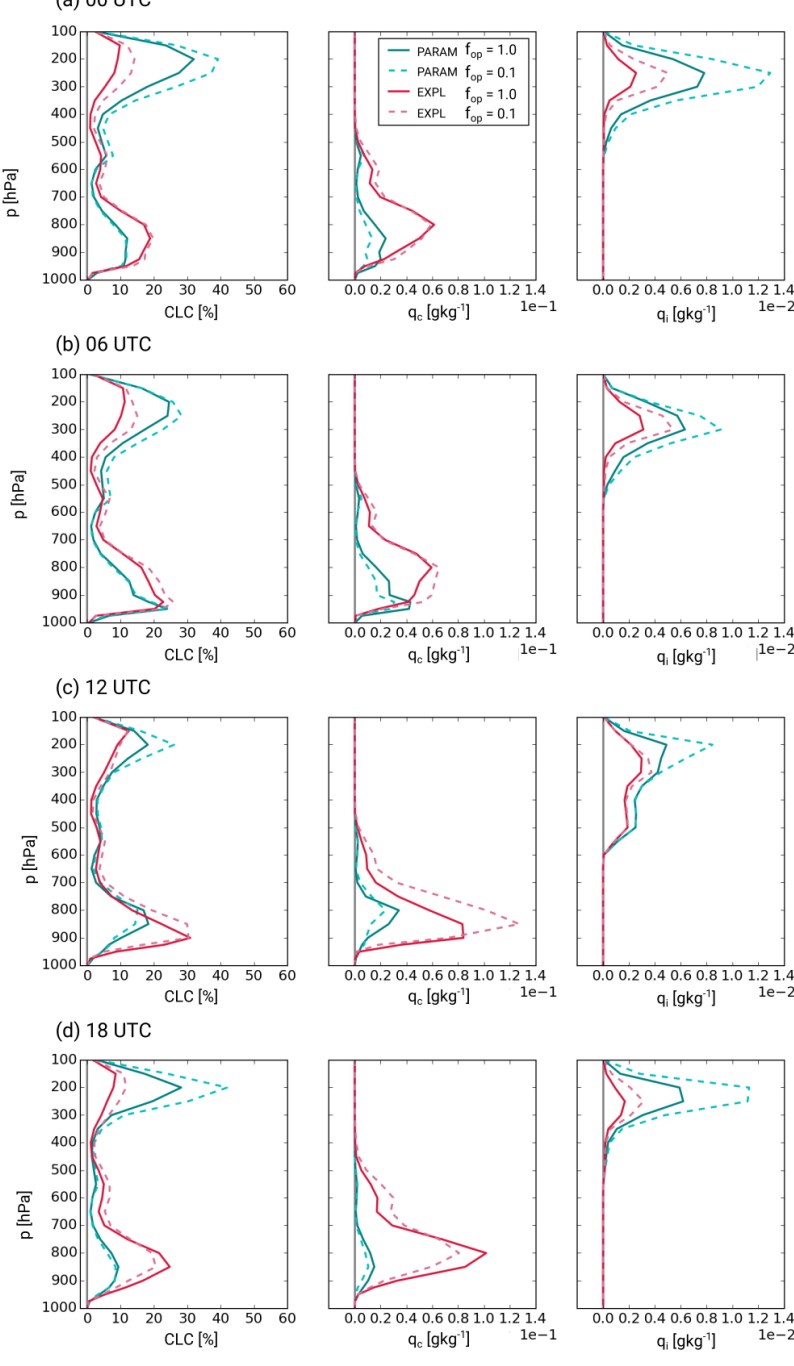

**Figure 6.** July 2006 mean profiles of $CLC$, $q_c$ and $q_i$ averaged over the DACCIWA box for experiments PARAM (green) and EXPL (red) and varying $f_{op}$ according to the legend at the top: (a) 00 UTC, (b) 06 UTC, (c) 12 UTC and (d) 18 UTC.





reach, respectively, more than 30% and almost 0.008 g kg$^{-1}$. This indicates a relationship of high-level clouds with the diurnal cycle of convection (Fig. 5) leading to an increase in the second half of the day.

Reducing the optical thickness of low clouds in PARAM ($f_{op} = 0.1$; dashed green lines in Fig. 6) has hardly any impact on low-level $CLC$ during nighttime but leads to a small decrease at 12 UTC and an even lesser decrease at 18 UTC, possibly
due to a deeper and/or drier PBL. Surprisingly, however, $q_c$ is decreased by on the order of 0.01 g kg$^{-1}$ at all times and most strongly so at 00 UTC, indicating that for $f_{op} = 0.1$ a similar cover of clouds is achieved with less liquid water. This aspect will be further discussed in the following subsection. For high clouds in contrast, both $CLC$ and $q_i$ increase markedly for all times with values on the order of 7% and 0.005 g kg$^{-1}$ at the peak of the profile at about 250 hPa. This is likely a reflection of the increased daytime convection in the sensitivity experiment, leading to more precipitation (Fig. 5) and generating substantially
more cirrus. This also suggests that part of the effect of more solar radiation reaching the surface through the optically thinned low clouds is compensated by an increase in high clouds. The comparison with the radiative transfer results by Hill et al. (2018) in the previous section, however, suggests that this is a relatively small effect overall.

Comparing the results for PARAM with those for EXPL reveals some substantial differences. Low clouds are more abundant in EXPL at all times, as already suspected in Sect. 3.2, contain substantially more liquid water and peak at 12 UTC rather than
06 UTC as in PARAM. $q_c$ can be up to 0.09 gkg$^{-1}$ higher for EXPL. The sensitivity of $q_c$ to $f_{op}$ has a much stronger diurnal cycle with little effect at 00 UTC, a small increase at 06 UTC, a large increase and deepening at 12 UTC followed by a decrease at 18 UTC (middle panels in Fig. 6). Consequently, the signals at 06 and 12 UTC go in the opposite direction in EXPL and than in PARAM. This rather unexpected results will be discussed in more detail in the following subsection. In addition there is a small increase in midlevel $q_c$ at all times. In stark contrast, high-level clouds are much reduced relative to PARAM in
both amount and $q_i$ at all times with values of up to 0.008 gkg$^{-1}$ lower in EXPL. However, the general sensitivity is similar for high clouds with an increase for $f_{op} = 0.1$ for all times. The magnitude again appears to be related to the diurnal cycle of convection, which is delayed in EXPL relative to PARAM (see Fig. 5). This comparison reveals that in many aspects the variations between EXPL and PARAM are larger than the differences between $f_{op}$ equals 0.1 and 1.0 for each experiment. To first order, the convective parametrization appears to transport moisture out of the low and midlevels to deposit it into the
convection-fed cirrus layer. This creates overall less sensitivity to our modifications of low clouds as already discussed in the context of Fig. 4 but also a weaker diurnal cycle in the sensitivities.

### 3.3.2 Vertical structure

Given the overall higher sensitivities and likely more realistic diurnal cycle in EXPL, we will begin the following discussion of thermodynamic changes with this experiment instead of PARAM. This discussion will help to shed more light into the low-
cloud behavior and sensitivities discussed in previous sections. Figure 7 shows DACCIWA box-averaged profiles of differences between the $f_{op} = 0.1$ sensitivity experiment and the $f_{op} = 1.0$ control run for $T$, specific humidity $q_v$, $RH$, turbulent kinetic energy $TKE$, $q_c$ and horizontal wind speed $v_{horiz}$. The colored lines show eight different times of day.

With respect to $T$ a relatively complicated vertical profile and diurnal cycle is found. Below 900 hPa, as expected, the reduced optical thickness of low clouds leads to more solar heating during the day and consequently an overall warming



**Figure 7.** Diurnal cycle (colored lines) of DACCIWA-box and July 2006 averaged profiles of differences $f_{op} = 0.1$ minus $f_{op} = 1.0$ for EXPL showing (a) $T$, (b) $q_v$, (c) $RH$, (d) $TKE$, (e) $q_c$ and (f) $v_{horiz}$.





peaking at 15 UTC with a slight cooling at 06 UTC (Fig. 7a). Immediately above that, there are indications of enhanced latent heat release within the low-level cloud deck, at least for some times of day when $CLC$ and $q_c$ increase (see Fig. 6) but during the day this effect is not clearly separable from the sensible heat fluxes in the PBL. Above that, around 725 hPa is a shallow layer with a slight cooling, most pronounced during the day and possibly due to radiative effects at the low-level cloud tops. The

increases in midlevel cloud and cloud water around 550 hPa (see Fig. 6) also leads to a warming below (latent plus radiative heating) and radiative cooling above, the latter most pronounced at nighttime. Finally, the cirrus layer peaking around 250 hPa also produces such a dipole pattern but with a much smaller diurnal cycle consistent with Fig. 6.

Signals in $q_v$ in contrast are much simpler and show a deep atmospheric moistening at all times (Fig. 7b). The only drying occurs in the lowest few hundred meters at 12 and 15 UTC, when substantial amounts of moisture are pumped into the elevated

low-level cloud layer where $q_v$ maximizes. An interesting time is 09 UTC, when $q_v$ is markedly enhanced near the surface. This may be related to an earlier start of the diurnal PBL growth (see discussion on $TKE$ below) or possibly also due to higher evapotranspiration in response to the increased precipitation (see Fig. 5). The second maximum in $q_v$ increase is found in the area of the midlevel cloud layer around 550 hPa. Due to generally low values in the cold upper-troposphere, changes in the cirrus layer are less evident in Fig. 7b. The net increase of column moisture and precipitation (Fig. 5) suggests a substantial

increase of moisture convergence into our study region. This will be further discussed in the next subsection. The signal in $RH$ (Fig. 7c) is a combination of the signals in $T$ and $q_v$. Given the large increases in $q_v$, $RH$ increases everywhere above 800 hPa at all times of day, with the profile reflecting some of the modulations in the area of the mid- and high-level cloud decks already discussed. Highest $RH$ increases of up to 5.5 % are found in the early morning, at the end of a period with convective moisture transports and radiative cooling. At the very lowest layers, the large increase in $T$, particularly during the day, leads

to a decrease in $RH$. The level with zero difference descends at night and ascends during daytime. It is lowest at 06 UTC, which facilitates the nocturnal low-level cloud formation for $f_{op} = 0.1$, leading to a slight increase in $CLC$ and $q_c$ (Fig. 6). At 12 UTC $RH$ near the surface is reduced but the higher values above 900 hPa help expanding the cloud deck upwards, while at 18 UTC the drying is so deep that clouds are reduced (cf. Fig. 6).

The discussion so far has illustrated the paramount importance of vertical mixing. To reveal the impact of low-cloud shielding

on turbulence, Fig. 7d shows the vertical profile of differences between $f_{op} = 0.1$ and $f_{op} = 1.0$ for $TKE$, which is increased at all levels and all times. Below 700 hPa turbulence gradually dies down from 18 UTC to 06 UTC. Due to the missing effect of low clouds in $f_{op} = 0.1$, $TKE$ differences increase markedly from 09 UTC to 15 UTC and rise upwards. 12 and 15 UTC show a secondary peak between 850 and 750 hPa, which is probably related to turbulence within the low-level cloud deck. Above 700 hPa, there is rapid increase from low values at 09 and 12 UTC to a maximum at 18 UTC, followed by a gradual decay. This

behavior clearly illustrates how deep convection communicates the – at first surface-based – signals into the entire troposphere. Finally, the localized maximum in $TKE$ differences around 900 hPa at night is an indication of a slightly enhanced NLLJ creating turbulence through shear (see Fig. 7f), which in turn helps the cloud formation.

Figure 7e shows the effect of the discussed changes in $RH$ and $TKE$ on $q_c$, shedding more light into the absolute values already discussed above (solid and dashed red lines in middle panels of Fig. 6). A good starting point to discuss the diurnal

cycle of this signal is 18 UTC, when the increase of deep convection is largest (Fig. 5) and creates more clouds above 750 hPa



and less in the main low-level cloud deck (Fig. 6d), as the deeper mixing reduces $RH$ (Fig. 7c). At 21 UTC the convective signal weakens and there are some first indications of increased $q_c$ in the nocturnal stratus deck around 925 hPa. As area-mean $RH$ is still negative at this level at this time (Fig. 7c), this is likely related to a greater variability within the box. The enhancement in $q_c$ in the low-level cloud deck increases and rises until 09 UTC. After 09 UTC the more dynamic evolution of

the daytime PBL in $f_{op} = 0.1$ leads to a more elevated low-level cloud deck containing more $q_c$ in the vertical column. This consists a negative feedback mechanism, as a (here enforced) reduction of low cloud leads to more cloud production, at least in the early part of the day.

  Finally, Fig. 7f shows impacts on horizontal winds. As already mentioned above, the $f_{op} = 0.1$ experiment has a stronger NLLJ developing around 18 UTC and lasting through the night. Only 12 and 15 UTC, when mixing is strongly increased (Fig.

7d), show a reduction of low-level wind speed. Above that, at the level of the African easterly jet (750–450 hPa) and at the level of the tropical easterly jet (300–150 hPa), $v_{horiz}$ is markedly decreased, a signal with a relatively small diurnal cycle. One possible explanation for this finding is a reduction of wind peaks through increased convective mixing, depositing more momentum in the layer of lower background winds at 400 hPa.

  Figure 8 shows the corresponding profiles for PARAM. Despite the overall consistent signal in rainfall and radiation as

documented in Fig. 4, there are many substantial differences between the two sets of experiments. The most striking probably is $TKE$ (Fig. 8d). Positive signals are restricted to the low levels during the day (09, 12 and 15 UTC) with the latter showing some indications for increased mixing reaching midlevels. All hours from 18 UTC to 09 UTC show decreased $TKE$ for most of the layer below 600 hPa and hardly any change at all above that. One needs to bear in mind, however, that the mixing through convection is not reflected in $TKE$ fields in PARAM and this is expected to increase. Nevertheless, the PARAM signals, at

least at low levels, are in clear contrast to EXPL (Fig. 7d) where vertical mixing increases everywhere. These differences are strong indicators that the interplay between PBL turbulence, shallow and deep convection fundamentally differs between the two model configurations. Particularly during nighttime PARAM shows a slight stabilization in the temperature profile (Fig. 8a) above 925 hPa that appears to suppress turbulence generation in this layer. This cooling may be related to the enhanced NLLJ (Fig. 8f) but it is not clear why this effect does not work in EXPL where an even more enhanced NLLJ and also a

stabilization is observed (Figs. 7a and f). The changes in mixing have profound impacts on many low-level fields, whereas more agreement between EXPL and PARAM is found at mid- and upper-levels, except for some changes in diurnal cycle. Despite a larger $SSI$ (see Fig. 4a), PARAM has a lower daytime increase in near-surface temperature, particularly at 15 and 18 UTC, suggesting a possible impact of the earlier triggering of convection in PARAM (see Fig. 5). Near surface $q_v$ (Fig. 8b) is strongly decreased at 09 UTC, probably due to the earlier onset of PBL mixing with transparent clouds, and then strongly

increased at 12 and 15 UTC, possibly due to the lack of deep mixing as in EXPL, leading to very large differences between the two sets of experiments. Combined, the changes in temperature and moisture lead to overall less pronounced changes in $RH$ at low levels (both negative near the surface and positive above; Fig. 8c), associated with mostly negative changes in $q_c$ (Fig. 8e) except for 09 UTC. These explain the somewhat unexpected results for $q_c$ discussed in the context of Figs. 7 and 6. In contrast to EXPL, PARAM operates a positive feedback mechanism, where a reduction in low cloud leads to a further reduction. This

may explain, why so many climate models show very large negative biases in cloud cover (Hannak et al., 2017). Overall this




PARAM

**Figure 8.** As Fig. 7 but for PARAM.

discussion demonstrates the enormous importance of vertical transport and mixing in a moist tropical environment where the PBL, low clouds and deep convection are closely coupled through radiative effects.





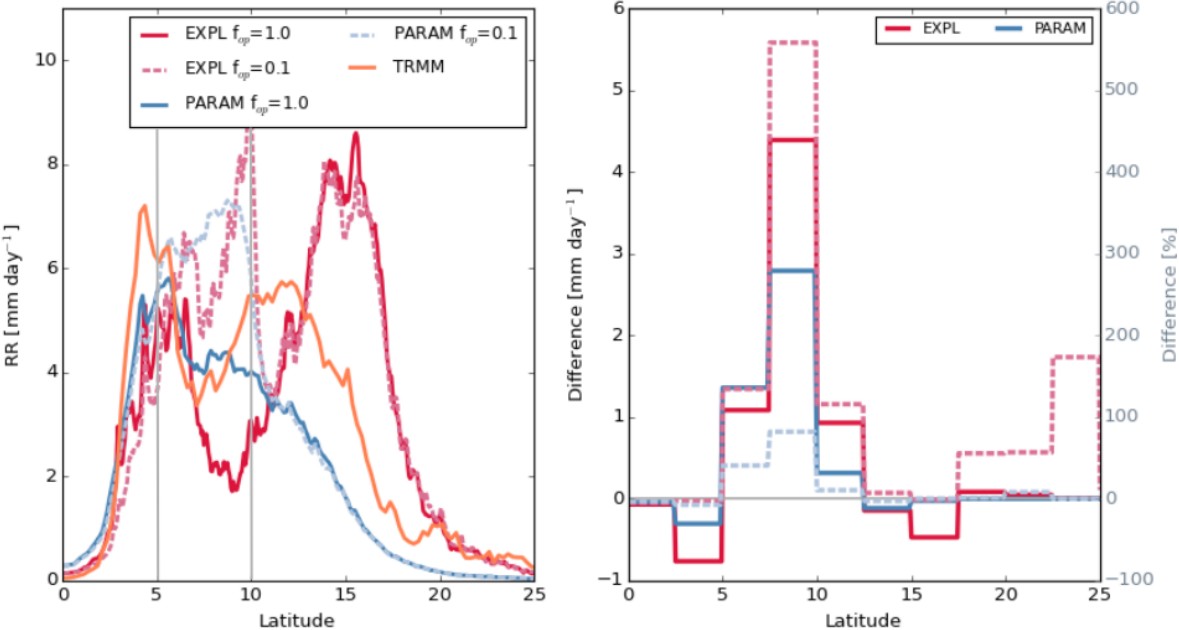

**Figure 9.** South–north distribution of $8°$W–$8°$E averaged $RR$ from various ICON simulations and TRMM observations (according to the legend) for July 2006: (a) absolute amounts and (b) differences $f_{op} = 0.1$ minus $f_{op} = 1.0$ as absolute (solid) and relative (dashed) values. For better visibility, the data points were binned every $2.5°$ latitude in (b).

## 3.4 Regional impact

### 3.4.1 Precipitation

The previous sections have revealed how moderate modifications to low-level cloudiness can profoundly change the diurnal cycle in many meteorological variables over southern West Africa, leading amongst other things to a substantial increase in

5    precipitation. This raises the question to what extent these modifications have an impact on neighboring regions or even on the entire WAM system. Does the increased precipitation over the DACCIWA box suppress precipitation to the north and south? Does this enhance or weaken the monsoon circulation?

Figure 9a shows zonally averaged ($8°$W–$8°$E) south–north distributions of precipitation for the ICON EXPL and PARAM experiments with $f_{op} = 0.1$ and $f_{op} = 1.0$ together with the corresponding TRMM observation, while Fig. 9b displays the

10   sensitivities in absolute and relative terms. As already discussed in the context of Fig. 2, the ICON control simulations show large deviations from TRMM. While the southern maximum around $5°$N, which is mostly related to the very moist Niger Delta region (see Fig. 2), is only slightly underestimated by both ICON runs, the Sahelian rainband is much too weak and shifted southward in ICON PARAM and too strong and shifted northwards in EXPL. The latter also shows a much wider



and drier gap between the rainfall maxima. Despite these differences, the response to reducing the cloud optical thickness is similar, with a large increase over the modification region itself (5–10°N) and immediately to the north, i.e. downstream with the monsoon flow, and rather small changes elsewhere. For PARAM differences outside of the DACCIWA box are small in both an absolute and relative sense (Fig. 9b). Largest differences occur in the northern half of the box with an increase of

almost 3 $\mathrm{mm\ day^{-1}}$ corresponding to about 80%. Changes in EXPL are generally more dramatic. Given the drier conditions over the Guinea Coastal region in the control run, the increase of almost 4.5 $\mathrm{mm\ day^{-1}}$ in the northern half of the DACCIWA box corresponds to an impressive 560%, while the southern half of the box and the 2.5°-strip to the north of it still reach increases on the order of 100%. To the north and south of that, small decreases in absolute values are found, most likely due to an immediate suppression by the enhanced convection in the box, but these are barely significant in a relative sense (Fig.

9b). Finally, to the north of 17.5°N there is a small increase in absolute values, which, given the increasingly dry conditions in this area, corresponds to considerable relative changes. This may suggest that modulations to the WAM allow a slightly deeper penetration of rainfalls into the continent but one month is probably too short to make any definite statements on this area.

### 3.4.2 WAM system

In order to better understand these precipitation signals, Fig. 10 shows corresponding south–north distributions of differences

between the two EXPL runs for various meteorological quantities and their diurnal variations. Despite the relatively small impacts on precipitation, it demonstrates that the influence of the low cloud manipulation is not restricted to the manipulated area itself (dark grey lines) but is transported northwards with the mean flow as proposed by Zheng et al. (1999). This is evident, for example, for temperature at 975 hPa, $T_{975}$ (Fig. 10a). The near-surface heating peaks at 15 UTC within the box reaching values well above 1.0 °C apart from the southernmost part, where inflow from the ocean creates cooling. Until 06

UTC the $T_{975}$ signal weakens in magnitude and drifts northward out of the DACCIWA box. This change in advection (possibly in addition to radiative changes) leads to an overall moderate warming of the 10–20°N strip with a maximum at the end of the night. Farther to the north, there is a moderate decrease in the afternoon, likely connected to the increase in rainfall in this area (see Fig. 9b). The very small $T_{975}$ decrease over the ocean could come from enhanced sensible heat fluxes over the cool coastal waters caused by stronger winds (see Fig. 10c).

The increase in low-level temperature and higher-level latent and radiative heating (see Fig. 7a) leads to a considerable decrease in surface pressure, $p_{sfc}$, peaking at 18 UTC with values of more than 0.6 hPa (Fig. 10b). This effect is clearly spreading downstream of the box as for $T_{975}$ (Fig. 10a) but also upstream, likely due to upper-tropospheric flow. Given the overall north–south pressure difference of the monsoon, this signal leads to a sharpening of the gradient near the coast and a weakening towards the Sahel. The change in pressure creates a marked signal in low-level circulation, represented here by

the meridional wind at 925 hPa, $v_{925}$ (Fig. 10c). Southerly winds into and within the box are enhanced by 1 $\mathrm{ms^{-1}}$ and more, particularly leading to an increased NLLJ, while the export towards the Sahel is reduced. This may explain the general tendency of a combined underestimation of low clouds and overestimation of NLLJ found for many climate models (Knippertz et al., 2011; Hannak et al., 2017). Wind signals generally tend to be smaller during the day when PBL turbulence creates a drag on the monsoon circulation (e.g. Parker et al., 2005; Marsham et al., 2013). These changes in circulation also explain the strong





Figure 10. South–north distribution of 8°W–8°E averaged differences of ICON EXPL $f_{op} = 0.1$ minus $f_{op} = 1.0$ for July 2006. Colored lines provide a 3-hourly resolution of the diurnal cycle of (a) $T$, (b) $p_{sfc}$, (c) meridional wind $v$ and (d) $q_v$. In (e) absolute $\theta_e$ curves and their difference are shown. Apart from $p_{sfc}$ and $T$ at 975 hPa, all variables are shown for 925 hPa.





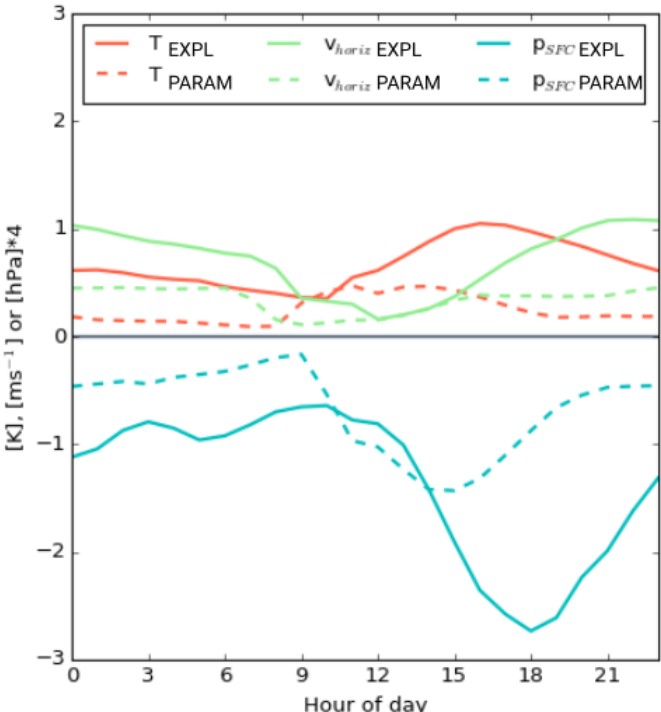

**Figure 11.** Maximum hourly value extracted from south–north distributions of July 2006 averaged diurnal cycles in Figs. 10 and 13 for $T$ and $v_{horiz}$ at 925 hPa as well as $p_{sfc}$ (according to legend).

moisture convergence into the DACCIWA box discussed above. In addition to the meridional component shown here, there is also strongly enhanced moisture convergence in the zonal flow in response to the reduced pressure (not shown). Enhanced evaporation due to stronger winds over the ocean (Fig. 10c) may also make a contribution. The link between temperature, pressure and wind is further illustrated in Fig. 11 that shows extrema in the south–north profiles of Figs. 10a–c per hour. $T_{975}$

5  signals clearly lag the diurnal cycle of solar radiation and peak around 16 UTC. Due to the additional effect of latent heating by convection, the $p_{sfc}$ minimum is reached with a delay of about 2 hours. Finally $v_{925}$ is even further delayed peaking around 22 UTC, when the increase in pressure gradient is still large but when daytime turbulence has died down.

The response in low-level moisture, represented here by $q_v$ at 925 hPa (Fig. 10d) shows a relatively complicated pattern. Signals within the DACCIWA box are predominantly positive, as already discussed, showing some signs of nocturnal advection

10  to the north similar to $T_{975}$ (Fig. 10a). Upstream over the ocean, $q_v$ is almost unchanged but downstream values are reduced almost everywhere at all times of day with largest differences during the night. This is unlikely a purely advective signal and is suspected to be partly caused by local vertical mixing. To further investigate this point, Fig. 12 shows vertical profiles of the $q_v$ signal at 00 and 12 UTC with the DACCIWA box and the 925-hPa level indicated by grey lines. At midnight (Fig. 12a), when





daytime convection dies down, a deep atmospheric moistening with values of up to $0.6\ \mathrm{gkg}^{-1}$ is found in the DACCIWA box and immediately to the north of it. The near-surface layer shows both positive and negative contributions. Upstream over the ocean moderate drying occurs in the 800 to 900 hPa layer, possibly related to enhanced subsidence in this area in response to the convective enhancement over land (this signal is clearly stronger at 00 UTC than at 12 UTC). The area to the north of the

DACCIWA box shows little signal above 700 hPa but an overall drying below with two local minima around 12 and 15°N. Where do these minima come from? A possible clue is provided by the signals at 12 UTC shown in Fig. 12b. The deeper mixing in the DACCIWA box with optically thinner low clouds creates an earlier PBL build-up, mixing moisture from lower to midlevels, as already discussed (see Fig. 7b). While southern areas in the DACCIWA box receive "fresh" moisture from the ocean, the low-level dry air is advected northward with the monsoon flow and reaches 12°N by 00 UTC (Fig. 12a) subject

to some vertical mixing. In the same way, the dry signal at 15°N at 00 UTC would originate in the DACCIWA box 36 hours earlier and the dry signal at 17.5°N at 12 UTC 48 hours earlier. An additional factor could be that the warmer low levels in this area (Fig. 10a) enhance vertical mixing and therefore entrainment of drier air into the PBL advected westward with the African easterly jet. Above this drier surface layer, the 12-UTC profile shows a moistening between 10 and 13°N that supports the idea of deeper mixing but possibly also some advection in the deep southerly monsoon flow. Through compensation, column

moisture does not change much in this zone and rainfall even increases (Fig. 9). From this discussion the observed small precipitation increase to the north of 17.5°N (Fig. 9) is not clear but a more detailed investigation is beyond the scope of this paper.

Finally, we would like to address the question of how the low clouds over southern West Africa affect the overall monsoon circulation. As mentioned in the Introduction, a well established conceptual model for this is the theoretical framework pro-

posed by Eltahir and Gong (1996), Zheng et al. (1999) and others, which relates the strength of the circulation to the large-scale meridional gradient in equivalent potential temperature $\theta_e$ within the PBL, assuming sufficient deep mixing by convection (e.g. Emmanuel, 1995; Nie et al., 2010). In order to apply this idea to our sensitivity experiments, Fig. 10e shows south–north distributions of $\theta_e$ at 925 hPa as absolute values (left) and as differences (right). As described by many studies, the monsoon is related to an enormous $\theta_e$ difference of almost 20 $^\circ C$ between the equator and about 12.5°N. Despite the large local im-

pacts discussed so far, our low-cloud modifications do not perturb this large-scale gradient significantly. Upstream changes are practically negligible. In the area of the $\theta_e$ maximum changes remain well below 0.5 $^\circ C$, resulting from the increase in temperature but decrease in low-level moisture. This is considerably smaller than observed interannual variations of 1–2 $^\circ C$ (e.g. Hurley and Boos, 2013) and consistent with the relatively small impact on precipitation in the Sahel evident from Fig. 9. In the DACCIWA box itself and immediately downstream, however, the combined increase of temperature and moisture

leads to $\theta_e$ changes of more than 1 $^\circ C$ and a strong local precipitation increase. This means that the reduction of the effective albedo over southern West Africa allows concentrating more energy and precipitation over land without the necessity of shifts between land areas. An interesting implication of this result is that whatever change in aerosol-radiation or -cloud interaction is caused through changes in anthropogenic emissions, it will likely have measurable local but probably no significant regional impacts.




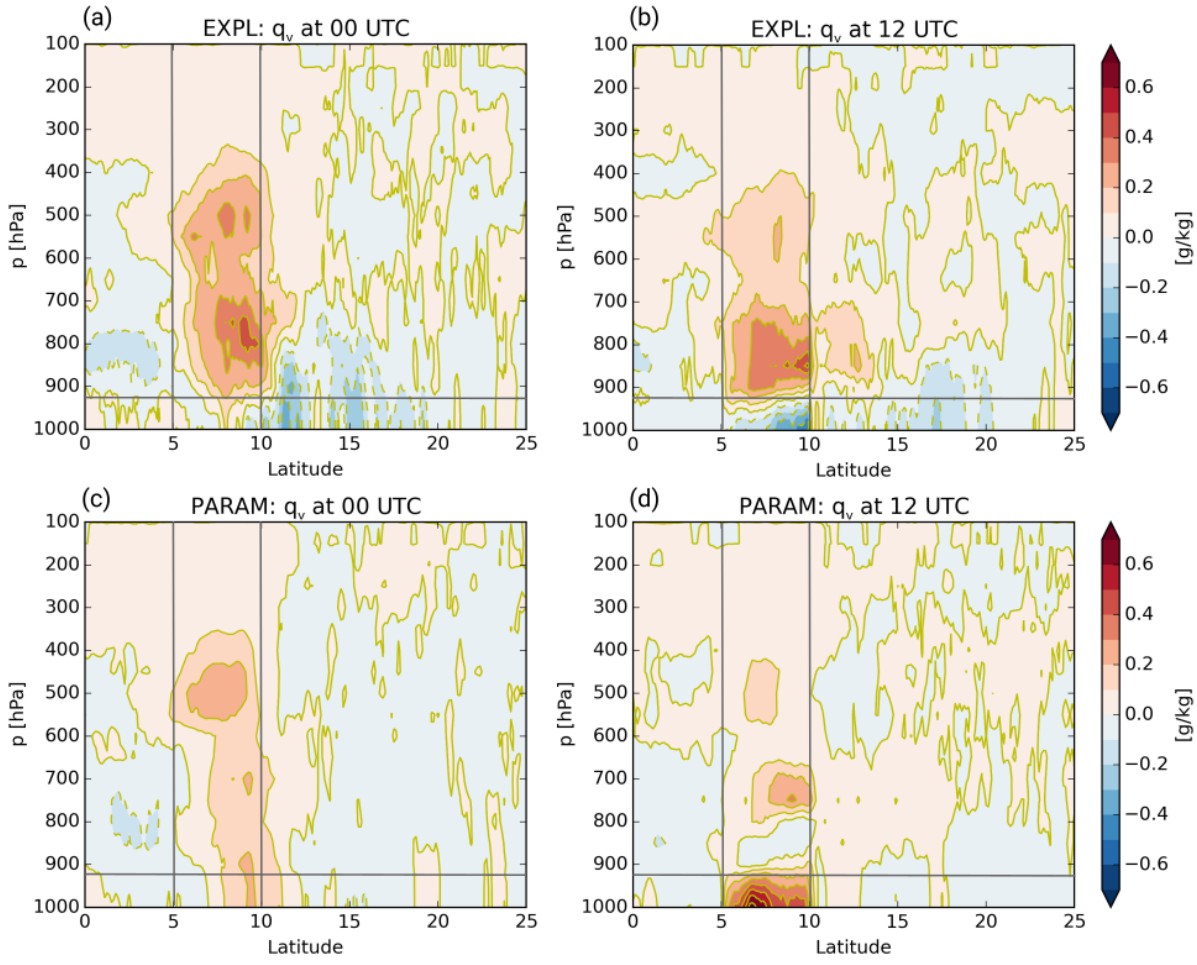

**Figure 12.** South–north distribution of 8°W–8°E averaged $q_v$ differences of ICON EXPL (top) and PARAM (bottom) $f_{op} = 0.1$ minus $f_{op} = 1.0$ for July 2006. Left panels show for 00 UTC and right panels 12 UTC. Grey lines indicate the borders of the DACCIWA box and the 925 hPa level used in Figs. 10 and 13.

For comparison, Fig. 13 shows the same fields as displayed in Fig. 10 but for PARAM. It clearly demonstrates the substantially smaller impact of reducing the optical thickness of low clouds on low-level fields within and beyond the DACCIWA box and the substantial changes to the diurnal cycle of the differences. The temperature signal (Fig. 13a) has a much smaller amplitude than in EXPL and peaks earlier in the day as discussed in the context of Fig. 8. Due to the main advection during the night, the impact on the Sahel is even further reduced and shows even a slight cooling during daytime. A similar behavior is found for $p_{sfc}$ (Fig. 13b) with a smaller and earlier peak and less impact on the Sahel than in EXPL. Given the relation of pressure and wind, it is no surprise to find a much reduced (or even reversed) signal in $v_{925}$ also (Fig. 13c). These differences are further illustrated in Fig. 11, showing the much flatter diurnal cycle in temperature with an earlier, less pronounced peak



**Figure 13.** As Fig. 10 but for PARAM.

already before midday (red curves). The pressure signal (blue curves) has a larger amplitude, as it is also related to latent

heating at upper levels but due to the different timings in precipitation (Fig. 5), a shift of 3 hours relative to EXPL is found.



With the pressure signal already much decreased around sunset, the wind response is weak (see also the discussion in Marsham et al., 2013) and shows very little diurnal variations (green curves).

With respect to $q_v$ at 925 hPa differences between PARAM and EXPL are again more complicated. There is more consistent drying over the ocean in PARAM but moistening with a similar magnitude compared to EXPL in the DACCIWA box (Fig. 13d),

however, with a much different diurnal cycle as discussed in the context of Fig. 8. Over the Sahel, the increase in meridional wind (Fig. 13c) during the night leads to a clearer signal of northward moisture advection in stark contrast to EXPL where substantial drying is found (Fig. 10d). Looking at the vertical structure of these signals (Fig. 12) underlines the paramount importance of vertical transports and mixing of moisture. PARAM has generally weak signals everywhere to the north of the DACCIWA box apart from the stronger low-level moisture advection at night just mentioned and does not show signs of the

diurnal pulses of dry advection discussed for EXPL above (Fig. 12c). Over the ocean to the south there is some agreement between PARAM and EXPL on a general drying of low- and midlevels. In the box itself, contrasts are extremely large at 12 UTC. At this time, EXPL shows effects of enhanced nighttime dry advection from the ocean at low levels and moisture left over from convective mixing from the previous day above 925 hPa (Fig. 12b). In PARAM convection is already active at this time, effectively removing tropospheric surplus and depositing it in the PBL (Fig. 12d). Due to the less effective vertical

transports during the day in PARAM, the moisture signal at midnight is substantially weaker in the free troposphere (cf. also Figs. 7b and 8b). These changes lead to an overall smaller signal in $\theta_e$ at 925 hPa within the DACCIWA box and to the north of it, too (Fig. 13e), apart from the $10 - 12°$N band, where nocturnal moisture advection is enhanced, as discussed above.

Finally, we also tested the time needed for the atmosphere to return to a normal state after a switch-off of the induced cloud changes in the model using the EXPL configuration. These experiments show that low-level variables such as surface radiation

and temperature react almost immediately to changes in low cloud during the day. Low-level cloud cover and rainfall respond after one full diurnal cycle, while impacts on higher and more remote regions can last days. More details can be found in the Supplementary Material.

In conclusion, this discussion shows that the parametrized treatment of convection not only affects the diurnal timing of precipitation but also impacts strongly on vertical mixing. Through a number of different mechanisms, these create substantial

differences in thermodynamic environments and ultimately in the sensitivity to modifications of low-level clouds, which is generally higher in EXPL than PARAM. The differences also impact on the propagation of signals to the Sahel in both magnitude and diurnal timings. Despite all this, precipitation signals are clearly dominated by the DACCIWA box itself with only minor impacts outside of the box.

# 4   Conclusions

In the present study, we analyzed the role of low-level clouds over southern West Africa on the local meteorology and larger monsoon system. These clouds play an important role in the energy budget and diurnal cycle during summertime and tend to be badly represented in many climate models (Hannak et al., 2017). They frequently form during the night close to the surface and often persist long into the following day. At their maximum diurnal extent, they cover a vast area of about $850\,000$ km$^2$ in





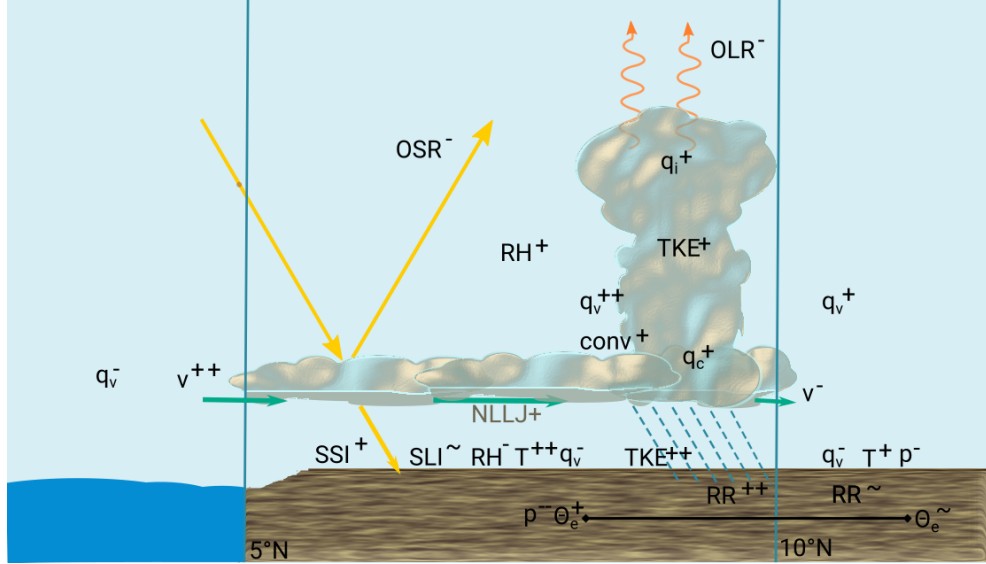

**Figure 14.** Conceptual sketch of the most important changes when reducing the optical thickness of low clouds based on the ICON EXPL experiments. For more details see Sect. 4.

southern West Africa (van der Linden et al., 2015). Their formation is linked to cold advection and turbulent mixing associated with the NLLJ and radiative cooling (Schrage and Fink, 2012; Schuster et al., 2013; Kalthoff et al., 2018). The role of these clouds in the WAM system was assessed here for the first time in a fully nonlinear way via sensitivity experiments using the ICON model from the DWD in NWP mode for July 2006. Cloud radiative effects were suppressed or enhanced in the model

over the main low-level stratus region 5–10°N and 8°W–8° by multiplying $q_c$ below 700 hPa with a constant factor $f_{op}$ before the call of the radiation scheme. Simulations with a horizontal grid-spacing of 13.2 km and parametrized moist convection (PARAM) were systematically compared to those with an additional nest over West Africa with a finer grid-spacing of 6.6 km and explicit convection (EXPL).

    Comparisons with ground and satellite-based observations of rainfall and radiation show substantial deviations between the

two model configurations and with the observations. PARAM reproduces the coastal rainfall maximum over the Niger Delta but struggles to represent the inland penetration of precipitation. It appears to have realistic $SSI$ but too much extinction of shortwave radiation in the atmosphere, leading to a negative bias in $OSR$. EXPL also reproduces the coastal rainfall well but in contrast to PARAM has a much too strong Sahelian rainband substantially further north than observed. EXPL appears to have slightly too many low clouds, leading to reduced $SSI$ and increased $OSR$. PARAM generally tends to have substantially

more high and less low clouds compared to EXPL. This demonstrates the enormous influence of convective parametrisation on the West African meteorology as already documented in Marsham et al. (2013). As both model configurations show marked disagreement with observations, a quantitative interpretation of the results appears questionable. However, we argue that we



can still use the model to investigate which sensitivities are robust and how convective parametrization modifies the sensitivity and the involved physical mechanisms.

Making low clouds more transparent to short- and longwave radiation creates a complicated atmospheric response. To summarize the main effects, Fig. 14 shows a schematic overview that reflects the changes found for EXPL, as this experiment

shows a more realistic diurnal cycle. Differences to PARAM will then be discussed below. Figure 14 concentrates on daily mean effects but at least for some parameters diurnal variations will be discussed, too. Note that in the NWP simulations $SST$s stay largely constant during the short run time. The southern and northern borders of the box with cloud modifications, i.e. 5 and 10°N, are marked by vertical lines in Fig. 14.

The first and most obvious aspect is that more transparent low clouds lead to more solar radiation reaching the ground

($SSI^+$) and less being reflected to space ($OSR^-$) during daytime. This leads to an increase in low-level temperature in the daily mean, but particularly during the afternoon ($T^{++}$). The associated decrease in stability triggers more turbulent mixing ($TKE^{++}$) and more deep convection ($conv^+$), leading to more convective mixing and a substantial increase in precipitation ($RR^{++}$). Particularly in the northern half of the modification region, rainfall increases by an impressive factor of 5! The almost logarithmic dependence of rainfall on $f_{op}$ illustrates the strong and dominating control the low clouds exert on the

triggering of convection. The increase in low-level temperature and free-tropospheric latent heating leads to a marked decrease in surface pressure, particularly around the convective peak at 18 UTC ($p^{--}$). This in turn sharpens the gradient to the south and creates and enhanced low-level jet over southern West Africa ($NLLJ^+$) and a stronger inflow from the Atlantic ($v^{++}$). At the same time, the export to the Sahel is somewhat reduced ($v^-$). This enhancement in meridional convergence concentrates moisture over southern West Africa and through the enhanced vertical mixing moistens the upper levels ($q_v{}^{++}$). As this largely

dominates over temperature effects, relative humidity ($RH^+$), cloud water ($q_c{}^+$) and cloud ice ($q_i{}^+$) are increased throughout the free troposphere. Only close to the surface and particularly during the day, the enhanced advection of dry air from the ocean and intensified mixing creating a deeper PBL, leads to lower absolute ($q_v{}^-$) and relative humidity ($RH^-$). At 18 UTC this also leads to less cloud cover and cloud water (not shown). At other times of day, the stronger NLLJ and the additional moisture lead to an increased cover and water content of low clouds, creating a negative feedback. Due to the increase in convection

and high clouds, less longwave radiation is emitted to space ($OLR^-$), while surface longwave effects are small due to the overall very moist and cloudy column ($SLI^\sim$). The latter is a strong indication of dynamic adjustments in the model. A recent study by Hill et al. (2018) estimated the effect of low clouds over southern West Africa from pure radiative transfer simulations on satellite-derived cloud data. While for the shortwave component (i.e. $SSI$ and $OSR$) both approaches point in the same direction, the longwave components are reversed.

Effects outside of the cloud modification box are substantially smaller (Fig. 14). Upstream over the ocean the most significant signal is a free tropospheric drying, possibly from enhanced subsidence related to the increase convection over adjacent land. Downstream over the Sahel, low-level advection with the southerly monsoon flow is a dominating effect. Despite the lower meridional wind speeds, the enhanced temperature and lower pressure from the south create impacts as far north as 20°N ($T^+$ and $p^-$). With respect to moisture, however, changes in southerly advection, low-level moisture content in the south and deeper

mixing caused by the higher near-surface temperatures lead to a drying of low levels ($q_v{}^-$), with a diurnal pulsing signature.





Above the PBL, however, some of the increased humidity in the south is advected towards the Sahel with the deep monsoon flow ($q_v{}^+$), leading to overall small changes in column moisture. Consistent with that and despite the many changes discussed, total rainfall over the Sahel is not strongly affected by the cloud modifications applied here ($RR^\sim$), apart from the immediate vicinity of the box. However, it is possible that the observed changes could still lead to differences in diurnal cycle and/or

organization of convection. Such effects can be found indeed in our model results, but the number of mesoscale convective systems per month is too small to draw any substantial conclusions from the modelled time period. The opposite signs of temperature and moisture changes over the Sahel lead to relatively small changes in low-level $\theta_e$ there ($\theta_e{}^\sim$), in contrast to southern West Africa, where $\theta_e$ is enhanced ($\theta_e{}^+$). An interesting implication of this is that the total magnitude of the north–south gradient in this quantity is not affected, which has been shown to be an important control of the overall monsoon

circulation (Eltahir and Gong, 1996; Zheng et al., 1999; Hurley and Boos, 2013). Therefore these results strongly suggest that errors or changes to low-level clouds over southern West Africa will likely have substantial local impacts but probably do not strongly affect neighboring regions, at least not in terms of rainfall.

A systematic comparison of the effects described for EXPL with the help of Fig. 14 reveals substantial differences when convective parametrization is used (PARAM). While the first-order effect on rainfall (strong increase over cloud modification

box and little impact elsewhere) is confirmed, differences in thermodynamic variables and the diurnal cycle are substantial. First of all, PARAM's diurnal cycle in rainfall is shifted forward by about 3 hours as in many models with parameterized convection (Marsham et al., 2013). This impacts on the sensitivity to low clouds in manifold ways. The low-level heating with more transparent clouds is reduced leading to reduced pressure and wind signals. Reduced and differently timed vertical mixing has large impacts on the diurnal cycle of the vertical distribution of moisture. This is most extreme at midday when PARAM

has a marked low-level increase of moisture with transparent low clouds, related to more convective rainfall, while EXPL has a marked decrease from stronger dry advection and PBL mixing. These differences lead to an overall decrease of low clouds and cloud water in PARAM in contrast to an increase in EXPL for most times of day. This unexpected positive feedback can serve as an explanation, why many models with convective parametrization show large negative biases in low-level cloud cover (Knippertz et al., 2011; Hannak et al., 2017). In addition, exports of temperature and moisture signals to the Sahel are reduced

and follow a different timing.

In conclusion, this study has for the first time demonstrated the enormous control of the persistent and widespread low clouds over southern West Africa on local rainfall, while impacts on neighbouring regions are moderate at best. These results suggest that the well documented low-cloud errors in many climate models (Hannak et al., 2017) can likely serve as an explanation for the often large precipitation errors in the Guinea Coastal region but not in the Sahel, a least not in terms of average amount.

Similar effects can be expected from changes in low-level aerosol, as already documented for a case study by Deetz et al. (2018a). Increases in aerosol optical thickness, e.g. through human activity, would therefore reduce precipitation in the region affected by the stratus. Such increases in anthropogenic activity have been observed and are projected to increase given the overall dynamic population and economic development (see Knippertz et al., 2015). It would be desirable to explicitly model this effect for longer periods using convection permitting resolution. A detailed treatment of aerosol processes, including wet

deposition and water uptake (Deetz et al., 2018b) will be required for a realistic representation of the problem. A suppression




of rainfall by aerosols could create a positive feedback by reducing wet removal. In addition, more work is needed to gauge the realism of the simulations used for this study. While comparisons with rainfall and radiation are presented here, it would be necessary to also evaluate low-level thermodynamic and dynamic fields. The recent DACCIWA field campaign (Flamant et al., 2018) has generated an exciting new dataset to make progress on this end, particularly through its extensive ground-
based measurements (Kalthoff et al., 2018). In the long run, it is hoped that these activities can improve weather and climate models over this crucial and densely populated region, as there is no hope to realistically model the local meteorology without a realistic representation of the diurnal behaviour of low clouds.

*Competing interests.*   The authors declare no competing interests.

*Acknowledgements.*  The research presented in this article has received funding from the European Union 7th Framework Programme
(FP7/2007-2013) under Grant Agreement 603502 (EU project DACCIWA: Dynamics–Aerosol–Chemistry–Cloud Interactions in West Africa).
Radiation measurements at Cotonou and Parakou were carried out by the IMPETUS project funded by the BMBF project IMPETUS (BMBF
Grant 01LW06001A, North Rhine-Westphalia Grant 313-21200200). We particularly thank Orou Goura Doussi and Michael Christoph for
their maintenance and data retrieval commitment for the Parakou station. We wish to thank Abdourahamane Konaré, Adama Diawara, and
Fidèle Yoroba for providing the radiation data from the Lamto Geophysical Observatory in Ivory Coast. The dataset SARAH from EUMET-
SAT's Satellite Application Facility on Climate Monitoring was used to evaluate the ICON model, as well as CERES and TRMM data from
NASA.



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
