# Peer review of "The role of low-level clouds in the West African monsoon system"

_Atmospheric Chemistry and Physics, 2018_

## Referee Comment (RC1) · R. Vogel (Referee) · 11 Oct 2018

Summary:

This study investigates the influence of low clouds on the West African monsoon system by performing various sensitivity experiments with the ICON model. The authors show that a decrease in low cloud optical thickness leads to a strong local increase in rainfall through associated changes in the diurnal cycle of convection. By contrasting simulations with parameterized and explicit convection, they make out important differences in the sensitivity of convection to the applied low cloud modification.

Recommendation: Minor revisions

Assessment:

[Figure]

This is a nice paper that fits well into the scope of ACP and its DACCIWA special issue. The manuscript is well structured and presents novel findings that support the notion that the misrepresentation of low-level cloudiness in climate models with parameterized convection likely leads to biases in precipitation and the thermodynamic structure in Southern West Africa. I list some general comments below, as well as a number of detailed comments and typographic suggestions that I would like the authors to address before the manuscript is published.
* * *
General comments:

1. Distinction of parameterization influence and resolution influence:

At the end of Section 2.2.2, I was missing a discussion of the influence of the changing resolution between the PARAM and EXPL experiments. Only on p.13, L6 you mention that "differences between PARAM and EXPL in Fig. 4 illustrate the sensitivity of the response to horizontal resolution and the use of convective parameterization", but everywhere else you neglect the potential influence of the changing resolution on the results. Marsham et al. 2013 isolated the influence of the convective parameterization by comparing experiments with 12kmPARAM, 12kmEXP and 4kmEXP. I'd suggest you refer to their study noting that the most important differences between the experiments are due to the convective parameterization, and that the increasing resolution between the experiments with explicit convection merely leads to quantitative differences. It's of course a bit trickier than that, but I think you wouldn't need to go in much more detail.

2. Negative & positive cloud feedbacks:

On p. 19, L6&L34 I stumbled over the sentences referring to the negative and positive low-cloud feedbacks. The way it is written, one thinks that you actually enforced a reduction in low cloud, rather than just a change in their opacity. By making the low clouds less opaque, you just manipulate their radiative effect, but e.g. not their effect

on the moisture budget or the microphysics. This should be made clearer.

3. Influence of organization of convection:

I would like to see some more discussion about the influence of changes in the organization of convection on the results. From Marsham et al. 2013 I take that mesoscale convective systems and the associated storm outflows are a significant component of the WAM system and I assume that they will also affect differences between your PARAM and EXPL simulations. On p.31, L5 you mention that you find effects of convective organization in your simulations. I understand that a detailed analysis of the role of convective organization would be beyond the scope of the manuscript, but maybe you can already appreciate some of the differences by looking at profiles of moisture variance and their diurnal cycle (similar to Figure 7). This might also be important for radiatively-driven secondary circulations that likely contribute to organizing the convection. I'd be surprised if changes in convective organization wouldn't be important in your experiments.

4. Use of commas:

I'm not a punctuation-expert, but I feel that there is a strong lack of commas throughout the manuscript. This distorts the flow and rhythm of reading. Examples are: p.3, L2 (season, low-level); p.4, L2 ((Sect 2.1), followed); p.6, L21 (set, ICON); p.7, L11 (given, concentrating); p.8, L8 (box, area-averaged); p.12, L9 ([...] Figure 4a), ranging); p.13, L32 (EXPL, but); p.16, L14 (Sect 3.2, contain); p. 25, L26 (maximum, changes); p.28, L3 (hPa, differences).

———————

More detailed comments:

p.1, L20-23: You should be a bit more specific here. Interactions of the WAM with the land surface? Representation of the hydrological cycle in West Africa?

p.2, L13: I don't understand what you mean with low-level processes. Do you mean

boundary-layer processes or land-atmosphere interactions? Or do you already refer to the local factors and surface characteristics that are the topic of the next paragraph?

p.3, L4: What does "this phenomenon" refer to here? The low-level stratus or the NLLJ?

p.3, L7: I would remove the details of the radiative transfer model ('using the two-stream radiative transfer model SOCRATES (Suite Of Community RAdiative Transfer codes based on Edwards and Slingo; Edwards and Slingo, 1996)')

p.3, L13: What do you mean with "but feedbacks were not considered explicitly"? Where they not represented, or not analysed? Please clarify.

p.6, L13: Maybe again refer to Figure 1 here.

p.7, L2: You haven't explicitly mentioned the control experiment yet. Maybe add a sentence on p.6, L19, saying that "f_op=1 corresponds to the control experiment."

p.8, Figure 2: I would suggest a few changes in this figure. I'd recommend using a white background for the maps and a different colour scale (e.g. the 'YlGnBu' palette from https://betterfigures.org/2015/06/23/picking-a-colour-scale-for-scientific-graphics/). Furthermore, the DACCIWA box could be shown in every panel.

p.9, L33: I don't really agree with the conclusion that ICON PARAM looks more consistent with the observations and ICON EXPL less. Together with CERES, ICON EXPL has a very good agreement with the few surface observations. This is in clear contrast to ICON PARAM, which tends to overestimate SSI compared to the surface observations. Maybe you can provide a more balanced conclusion of this paragraph.

p.9, L27-L29: Please clarify the sentence "brighter than the surface (except for snow) but in this region is likely still contaminated by clouds."

p.12, L32: Depth of cloud modification layer: I thought you were modifying clouds

below 700 hPa rather than below 750 hPa (see p.6, L13).

p.14, Figure 5: This figure has a relatively wild mix of colours and line types. Where applicable, I'd suggest to use more consistent colours throughout the paper, e.g. greenish colours for PARAM and reddish for EXPL (as in Figure 6). Further, I'd restrict the use of dashed lines in Figure 5 to the simulations with f_op=0.1.

p.16, L24-26: I don't really understand what you want to say here. The convective parameterization is by design responsible for vertical moisture transport. But also explicit convection transports moisture in the vertical. So I don't understand how this would explain the lower sensitivity. Do you want to say that "the convective parameterization more efficiently transports moisture [...] compared to explicit convection."?

p. 19, L18 onwards: I don't know exactly how TKE is treated in the parameterization, but as you say that the "mixing through convection is not reflected in TKE fields in PARAM", it's not surprising that the TKE profiles are very different. For me, the most striking difference between Figure 7 & 8 instead lies in the qc profiles. I would suggest some restructuring of this paragraph. It was also not always clear to me whether you are comparing PARAM and EXPL or the response to the opacity change for PARAM. This should be clarified.

p.21, L3: You didn't state the sign of the modification of low clouds, but then say that it leads to substantial increases in precipitation. I'd suggest to reformulate the sentence as follows: "[..] how moderate reductions in low-cloud opacity [...]".

p.22, L7: I wouldn't use the word "impressive" here, especially as you stress in other parts of the manuscript that a quantitative interpretation of the results is questionable. Maybe just use "an increase of 560%". The same is true for p.30, L13 ("an impressive factor of 5!"). I also don't really like the use of the word 'enormous' (e.g. p.25, L24; or p.31, L26), but that might be a matter of taste.

p.22, L31: I don't see how the sentence "This may explain the general tendency..." fits

in the discussion of the EXPL simulation here, as I assume that this might be different between explicit and parameterized convection.

p.27, Figure 13: Change legend in panel (e) to f_op=1.0 & f_op=0.1.

p.28, L 13-14: I don't understand what you mean with "effectively removing tropospheric surplus and depositing...", maybe something is missing here?

p.30, L21: I would assume that air advected from the ocean is moist, not dry. Am I missing something here?

S1, p.1, L27-29 and Figure S1: maybe add a measure of spread between the different runs to indicate the variability.

—————————

Typographic suggestions:

p.2, L7: and ITD shift –> and the ITD shift

p.2, L14: Eltahier –> Eltahir

p.3, L4: Omit either realistically or correctly.

p.5, L23: allows –> allow

p.5, L24: terrain following –> terrain-following

p.6, L21: remove grid ("a grid spacing of 13.2 km grid...")

p.7, L3: first (Sect. 3.1) –> first section (Sect. 3.1)

p.7, L17-L18: add a "the" in front of "adjacent ... highlands"

p.9, L1: "by on the order of" –> by about

p.12, L18: from –> of

p.13, L21: following –> followed

p.14, L4: clod –> cloud

p.16, L5: by on the order –> by about

p.16, L8: with values –> with absolute increases

p.16, L18: remove "than"

p.16, L18: results –> result

p.19, L18: hardly any change at all above –> hardly any change above

p.22, L26: with values –> with decreases

p.25, L32: aerosol-radiation or –cloud interaction –> aerosol-radiation or aerosol-cloud interaction

p.28, L21: impacts on higher and –> impacts on higher levels and

---

## Referee Comment (RC2) · Anonymous Referee #1 · 12 Oct 2018

This study uses the ICON model to investigate the influence of low level clouds over southern West African on the monsoon system. The authors find that in the perturbation region, precipitation depends strongly on the optical thickness of the low clouds. When representing convection explicitly, the model was even more sensitive to the low level cloud thickness. Downstream of the perturbed region there is very little effect on precipitation, due to temperature and moisture having opposing signals. The manuscript is well written and within the scope of ACP and the DACCIWA special issue. I recommend publishing this work after minor revisions and addressing reviewer comments.

Major Comments:

While the explanation of how the clouds were altered in the model was very clear and

innovative, I have several comments about the general setup of the model and how the simulations were performed and analyzed.

First, why was the ICON model used, and more specifically, why was a global model used for this experimental setup? For such short runs, I would think a regional or at least a nested global model would be sufficient. You state that computational cost limited your runs.

Do you used fixed SSTs or an interactive ocean (or ocean surface)? Since the simulations are so short, I don't expect the treatment of the ocean surface to have much effect, but it would be good to know.

For the EXPL runs, what is the domain of the nested grid? This is probably unimportant, but could have implications as odd things can happen on the boundary between nested grids.

Finally, what period is the data analysis averaged over? Is it the final four days of the five day simulations? This would be congruent with five day simulations having a single day overlap when the model was started every fourth day, however, it should be clearly stated. Is this period the same for analysis of both the local and regional response? The timescale of the local and regional response is an important factor in interpreting your results, so you should clearly state your averaging period for analysis and support your choice with evidence from your simulations (as in the supplementary material) and/or the literature.

When describing the ICON model base state and comparing it to obs and reanalyses it may be helpful to also know how the control simulations compare with the obs/reanalyses/Hannak et al. In terms of cloud fraction and LWP/IWP. Is it a model that produces a reasonable amount of low level clouds in the base state? The discussion on pages 8 and 9 somewhat address this, but since the paper is on clouds, it might be nice to just state how the control simulations cloud fields compare to obs/reanalyses.

The entire paragraph at the end of Page 12 and beginning of Page 13 is a bit confusing and unclear. (Page 12, Line 26 through Page 13, Line 5) Why do you connect the discrepancy in the longwave component with a dynamical response, but not the discrepancy in SSI? Could the change in upper-level clouds account for this difference? It is important to point out that July 2006 might not have been the most average year, but I'm a bit perplexed by why that can explain SSI and not OLR. Maybe I'm missing something here.

Page 19, Line 15: "most striking" is a bit subjective, don't you think? I agree that it is very striking, but I was immediately more intrigued by the low level qv and qc, whose signals can be interpreted as due to changes in vertical mixing, some of which can be explained by TKE. I think describing and explaining the signals in this manner might be more causal, but this, I suppose, is more personal opinion. This is also a massive paragraph and could probably be broken up between these two features.

The first panel in Figure 9 is really fantastic! However, I feel that the aspect of it that discusses the differences between ICON and TRMM as well as the description in the text (Page 21, Line 10 through Page 22, Line 1 "As already discussed . . . gap between the rainfall maxima") should be moved to Section 3.1 where you are discussing the differences in the control simulations and the TRMM. When inserted here it subtracts from the main point of the section which is to show how the perturbation effect precipitation into the Sahel.

Page 22, Line 11: "This may suggest that modulations to the WAM allow a slightly deeper penetration of rainfall into the continent but one month is probably too short to make any definite statements on this area." I am a bit confused by this statement. You are suggesting that only looking at July 2006 might not be enough to make a robust statement?

Page 25, Lines 32-34: "An interesting implication . . . no significant regional impacts." This statement makes it sound like you are generalizing your result to other regions,

how do you support that claim? Or are you focusing on the response to just altering the clouds in the DACCIWA region? Your statement in the summary (beginning Page 31, Line 10 with "Therefore these results . . .") is much more well supported.

Page 28, Lines 18-22: This paragraph feels out of place. Why are you concerned with how quickly the atmosphere returns to its normal state? The statement "impacts on higher and more remote regions can last days" is misleading as the supplemental material suggests that the residual impacts on remote regions are complicated by the chaotic nature of the atmosphere. I think the plot in the supplemental material is quite interesting and helps explore the timescales of the response, as well as the persistence of changes when the altered clouds are removed. It might be useful to plot the envelop of the anomalies in Figure S1, in order to show the variability between the ensemble members. This would help clarify what is a consistent response to the forcing and what is due to internal variability.

What is the remote (Sahel) response of precipitation for the positive values of f_op = 3 and 10 in the PARAM experiments? I realize these are extreme values, but it would be interesting to see if the same signals appear.

Minor Comments & Typographic Corrections:

Many passages would be improved by the introduction of commas, especially the Oxford comma. I've identified a few below, but if you'd like, I would be glad to mark up the submitted manuscript with where I believe commas would help separate clauses and help clarify the manuscript.

Page 1, Line 7: effect

Page 1, Line 24: "do not show skillful forecasts of precipitation for the next days", is awkward, perhaps "do not produce skillful short term precipitation forecasts" is clearer.

Page 2, Line 9: "determined by the WAM system", is unclear, what about the WAM system determines these characteristics? "are connected within the WAM system"

[Figure]

emphasizes the challenge of the interconnectedness of the WAM.

Page 2, Line 32: I would remove "which is currently gaining increasing attention".

Page 3, Line 17: misrepresenting

Page 3, Lines 19-30: I would reorganize this entire paragraph to flow better.

This study is part of the Dynamics-Aerosol-Chemistry-Cloud Interactions in West Africa (DACCIWA) project (Knippertz et al., 2015) that aims to better understand the consequences of the rapid increase of anthropogenic emissions in West Africa on the local air quality, weather and climate. To the best of our knowledge, it is the first to analyze the radiative impact of the low-level cloudiness over southern West Africa on the thermodynamics and dynamics of the regional atmospheric system in a fully non-linear and systematic way. The analysis is based on a number of targeted sensitivity experiments using the numerical weather prediction model ICON (Icosahedral Nonhydrostatic), systematically changing the optical thickness of the model clouds. This allows us to clarify the impact of the inter-model spread in cloudiness found in Hannak et al. (2017) on the overall monsoon development in both parameterized and explicit regimes of convection. Although aerosols are not directly modeled in our experiments, the effects found for imposed changes of cloud optical thickness also help to understand variations in the natural system brought about by aerosol effects on cloud properties and radiation, which in a similar way control the amount of shortwave radiation reaching the surface or interact with clouds through modifications in the diurnal cycle of the PBL (e.g. Deetz et al., 2018a).

Page 6, Line 4: Green's

Page 6, Line 7: (Dee et al., 2011), and do not use data assimilation. (the comma helps separate the idea that the simulations are not using assimilation instead of ERA-I, which clearly does)

Page 6, Lines 10-11: Initializing ICON runs at 00 UTC would mean starting the runs

during the development phase of the low-level clouds and therefore the runs were initialized at 12 UTC. ("would mean to start directly" is awkward and "preferred" makes it unclear that the runs were in fact started at 12 UTC)

Figure 2: maybe outlining the box in each subplot would help highlight the similarities and differences

Page 9, Line 11: remove "Similar to Fig. 2" it's not needed

Figure 4: why is the axis dimension for SLI so large? All the other subplots are so well framed, but this one has so much empty space it looks weird. Maybe 400-430 W/mˆ2 would suffice?

Page 12, Line 6: How do the fully nonlinear processes represented in the ICON . . .

Page 12, Line 26: I feel this might read better as: "The simple linear model used by Hill et al. (2018) allows a rough estimate of how much of the change in the ICON radiative fluxes are due to direct . . . "

Page 12, Line 29: I think it would be more proper to not capitalize "Increases" after the colon, to signify that it is not a sentence fragment. Usually, you should only capitalize the first word after a colon if the clause is independent.

Page 16, Line 23: equal to

Figure 10: The legend for subplot (e) should have fop = 0.1 for the red curve.

Page 24, Line 4: south-north profiles of Figs. 10a-c for each hour

Page 26, Line 1: There needs to be a better transition here. Begin with a clear sentence that you are shifting back to looking at the PARAM simulations and why, before introducing the figure.

Page 28, Line 9: remove "just mentioned"

Section 4, First Paragraph: The second sentence introduces the idea of representation

of clouds in models, but then immediately returns to the idea of describing the clouds in reality. The paragraph might flow better this way:

In the present study, we analyzed the role of low-level clouds over southern West Africa on the local meteorology and larger monsoon system. They frequently form during the night close to the surface and often persist long into the following day. At their maximum diurnal extent, they cover a vast area of about 850 000 km2 in southern West Africa (van der Linden et al., 2015). Their formation is linked to cold advection and turbulent mixing associated with the NLLJ and radiative cooling (Schrage and Fink, 2012; Schuster et al., 2013; Kalthoff et al., 2018). These clouds play an important role in the energy budget and diurnal cycle during summertime and tend to be badly represented in many climate models (Hannak et al., 2017). The role of these clouds in the WAM system was assessed here for the first time in a fully nonlinear way via sensitivity experiments using the ICON model from the DWD in NWP mode for July 2006.

Supplemental Material, Page 1, Line 4: Would read better as: "To investigate this we use EXPL experiments, in which f_op = 0.1 is applied for the first 4 days . . ." The phrase "as in EXPL" makes it unclear what type of experiment is being done here.

---

## Author Response (AR1)

**Reply to referee #2, Raphaela Vogel**

Dear Raphaela,

We are grateful to you for the positive assessment of our study, for insightful remarks and for the valuable recommendations. We appreciate all the comments; we took them into account while preparing the revised version of the manuscript.

Below, the original comments are given in blue color.
The text added to the revised version of the manuscript is marked by red color.
* * *
General comments:
1. Distinction of parameterization influence and resolution influence:
At the end of Section 2.2.2, I was missing a discussion of the influence of the changing resolution between the PARAM and EXPL experiments. Only on p.13, L6 you mention that "differences between PARAM and EXPL in Fig. 4 illustrate the sensitivity of the response to horizontal resolution and the use of convective parameterization", but everywhere else you neglect the potential influence of the changing resolution on the results. Marsham et al. 2013 isolated the influence of the convective parameterization by comparing experiments with 12kmPARAM, 12kmEXP and 4kmEXP. I'd suggest you refer to their study noting that the most important differences between the experiments are due to the convective parameterization, and that the increasing resolution between the experiments with explicit convection merely leads to quantitative differences. It's of course a bit trickier than that, but I think you wouldn't need to go in much more detail.

This is a helpful suggestion. We were aware of this study, and the fact, that the differences found in this study were mainly quantitative for varying horizontal resolutions. The same became apparent for ICON in a small test case, which was not included in the paper. Therefore, we did not study it in more detail, but should have mentioned this in the article. We added the following sentences at the end of section 2.2.2:

> The same study differentiated between the effect of parametrization and the effect of horizontal resolution by comparing experiments with 12 km grid-spacing and both parametrized and explicit convection as well as explicit convection at 4 km. It was found that the dominating factor is the convective parametrization, which substantially alters the dynamics of the monsoon system, while the influence of the horizontal grid-spacing is mainly of quantitative nature. Building on these results, we will concentrate on differences between parametrized and explicit convection and pay less attention to resolution effects.
* * *
2. Negative & positive cloud feedbacks:
On p. 19, L6&L34 I stumbled over the sentences referring to the negative and positive low-cloud feedbacks. The way it is written, one thinks that you actually enforced a reduction in low cloud, rather than just a change in their opacity. By making the low clouds less opaque, you just manipulate their radiative effect, but e.g. not their effect on the moisture budget or the microphysics. This should be made clearer.

We tried to explain this better by adding the following lines:

This denotes a negative feedback mechanism, as a (here enforced) reduction of low cloud opacity leads to more cloud production, at least in the early part of the day. Recall that the modification was only applied to the cloud optical thickness, as seen by the radiation scheme.
* * *
3. Influence of organization of convection:
I would like to see some more discussion about the influence of changes in the organization of convection on the results. From Marsham et al. 2013 I take that mesoscale convective systems and the associated storm outflows are a significant component of the WAM system and I assume that they will also affect differences between your PARAM and EXPL simulations. On p.31, L5 you mention that you find effects of convective organization in your simulations. I understand that a detailed analysis of the role of convective organization would be beyond the scope of the manuscript, but maybe you can already appreciate some of the differences by looking at profiles of moisture variance and their diurnal cycle (similar to Figure 7). This might also be important for radiatively-driven secondary circulations that likely contribute to organizing the convection. I'd be surprised if changes in convective organization wouldn't be important in your experiments.

[Figure]

More organised convection would quite likely lead to more extreme values in q_v (i.e. drier in some regions and enhanced concentration of moisture in others), therefore the q_v variance in EXPL should be greater than in PARAM. The figure above displays the average differences in variance (EXPL – PARAM). What one can see here is an always positive difference with some diurnal cycle. This confirms the assumption of higher organisation. But this short analysis is too indirect in our eyes to be included in the article. Therefore we added on p14l24:

The latter is reflected in a larger variance of q_v throughout the lower and mid-troposphere in EXPL than in PARAM (not shown).
* * *
4. Use of commas:
I'm not a punctuation-expert, but I feel that there is a strong lack of commas throughout the manuscript. This distorts the flow and rhythm of reading. Examples are: p.3, L2 (season, low-level); p.4, L2 ((Sect 2.1), followed); p.6, L21 (set, ICON); p.7, L11 (given, concentrating); p.8, L8 (box, area-averaged); p.12, L9 ([...] Figure 4a), ranging); p.13, L32 (EXPL, but); p.16, L14 (Sect 3.2, contain); p. 25, L26 (maximum, changes); p.28, L3 (hPa, differences).

We tried our best to improve the punctuation throughout the paper, including your specific suggestions.
* * *
* * *
More detailed comments:
p.1, L20-23: You should be a bit more specific here. Interactions of the WAM with the land surface? Representation of the hydrological cycle in West Africa?

The first statement is explained by the following two sentences, which enumerate some of the difficulties in modeling of the WAM system. We modified the first sentences to the following:

Modelling the West African monsoon (WAM) system is a challenge, as reflected for example in large disagreement in rainfall, surface air temperature and cloud cover between models participating in the Coupled Model Intercomparison Project phase 5 (CMIP5) (Roehrig et al., 2013). Climate and weather models show a considerable inter-model spread when studying for example the influence of sea surface temperatures (SSTs) on the WAM circulation (Xue et al., 2010; 2016; Rodriguez-Fonseca at al., 2015), interactions of the WAM with the land surface (Boone et al., 2009) or the representation of the hydrological cycle in West Africa (Meynadier et al., 2010; Poan et al., 2016).
* * *
p.2, L13: I don't understand what you mean with low-level processes. Do you mean boundary-layer processes or land-atmosphere interactions? Or do you already refer to the local factors and surface characteristics that are the topic of the next paragraph?

We added:
Several studies stress the importance of low-level processes, such as near-surface moisture advection or turbulent fluxes for the development of the WAM (Perill´e et al., 2016; Eltahir and Gong, 1996).
* * *
p.3, L4: What does "this phenomenon" refer to here? The low-level stratus or the NLLJ?

The evolution of the low-level stratus, which is connected to the low-level jet. We added:

Climate models struggle to realistically represent the diurnal cycle of the stratus in terms of cloud amount and occurrence as well as wind speed (Knippertz et al., 2011; Hannak et al., 2017).
* * *
Since the results can depend on the radiative transfer method that was used, we would like to keep this information. A delta-two-stream method could not account for radiative effects of cloud edges, for example.
* * *
The sentence was modified to read "analysed" instead of considered.
* * *
Done.
* * *
Thank you! We added on p6,l29:

f_op is varied from 0.1 to 10, where f_op=1 corresponds to the control experiment.
* * *
We modified the figure as suggested.
* * *
You are right but ICON PARAM agrees better with the observations in terms of area-averaged solar surface irradiance, as explained in the respective paragraph:
"ICON EXPL shows the lowest SSI values with an area average of 164.7 W m$^{-2}$ (Fig.\ \ref{solclim}a), much lower than PARAM with 191.6 W m$^{-2}$ (Fig.\ \ref{solclim}b).
….
Evaluating this with observations is a challenge due to the many assumptions made in satellite-derived SSI and the few surface observations.
…..
In contrast, CERES does not seem to suffer from this problem due to a

different retrieval strategy (Fig.\ \ref{solclim}d). The box-averaged SSI is 188.4 Wm$^{-2}$ and therefore very close to the ICON PARAM value, although with much less fine structure."

Therefore we change only the last sentence to clarify this:

Overall this analysis demonstrates a significant observational uncertainty and suggests an overestimation of clouds in ICON EXPL leading to low average $SSI$, while ICON PARAM fields are more consistent with observations in this regard.
* * *
p.9, L27-L29: Please clarify the sentence "brighter than the surface (except for snow) but in this region is likely still contaminated by clouds."

Due to the high reflection, cloudy pixels appear brighter than cloud-free pixels for a satellite, therefore the surface albedo can be determined from the lowest irradiance measurements. In the DACCIWA region, unfortunately, it is not easy to find a cloud-free irradiance in a specific pixel (usually one month of measurements is used to find the surface albedo, this has to be done for each pixel and each hour of day). We modified the explanation to make this clearer:

As cloudy pixels appear brighter than cloud-free ones for SEVIRI, the surface albedo is estimated from the lowest irradiance measurement found per pixel in a given time period. In SWA, however, it is often difficult to find cloud-free scenes, leading to an overestimation of surface albedo (see also discussion of this problem in Hannak et al., 2017).
* * *
p.12, L32: Depth of cloud modification layer: I thought you were modifying clouds below 700 hPa rather than below 750 hPa (see p.6, L13).

Well spotted. We corrected this to 700 hPa instead of 750.
* * *
p.14, Figure 5: This figure has a relatively wild mix of colours and line types. Where applicable, I'd suggest to use more consistent colours throughout the paper, e.g. greenish colours for PARAM and reddish for EXPL (as in Figure 6). Further, I'd restrict the use of dashed lines in Figure 5 to the simulations with f_op=0.1.

We modified the Figure to be more clear and updated the figure caption accordingly.
* * *
p.16, L24-26: I don't really understand what you want to say here. The convective parameterization is by design responsible for vertical moisture transport. But also explicit convection transports moisture in the vertical. So I don't understand how this would explain the lower sensitivity. Do you want to say that "the convective parametrization more efficiently transports moisture [...] compared to explicit convection."?

The parametrization scheme transports more moisture, than the explicit convection does. We modified the sentence to read:

To first order, the convective parametrization appears to transport moisture more efficiently out of the low- and mid-levels to deposit it into the convection-fed cirrus layer, as compared to explicit convection.
* * *
p. 19, L18 onwards: I don't know exactly how TKE is treated in the parameterization, but as you say that the "mixing through convection is not reflected in TKE fields in PARAM", it's not surprising that the TKE profiles are very different. For me, the most striking difference between Figure 7 & 8 instead lies in the qc profiles. I would suggest some restructuring of this paragraph. It was also not always clear to me whether you are comparing PARAM and EXPL or the response to the opacity change for PARAM. This should be clarified.

We agree, and reviewer 1 commented on the same paragraph that was admittedly not easy to read. We restructured it as such (this time in green):

Figure 8 shows the corresponding profiles for PARAM. Despite the overall consistent signal in rainfall and radiation as documented in Fig. 4, there are many substantial differences between the two sets of experiments.

Despite a larger SSI (see Fig. 4a), PARAM has a lower daytime increase in near-surface temperature, particularly at 15 and 18 UTC, suggesting a possible impact of the earlier triggering of convection in PARAM (see Fig. 5). Near surface $q_v$ (Fig. 8b) is strongly decreased at 09 UTC, probably due to the earlier onset of PBL mixing with transparent clouds, and then strongly increased at 12 and 15 UTC, possibly due to the lack of deep mixing as in EXPL, leading to very large differences between the two sets of experiments. Combined, the changes in temperature and moisture lead to overall less pronounced changes in RH at low levels (both negative near the surface and positive above; Fig. 8c), associated with mostly negative changes in $q_c$ (Fig. 8e) except for 09 UTC. These explain the somewhat unexpected results for $q_c$ discussed in the context of Figs. 7 and 6. In contrast to EXPL, PARAM operates a positive feedback mechanism, where a reduction in low cloud leads to a further reduction. This may clarify, why so many climate models show very large negative biases in cloud cover (Hannak et al., 2017).

Increased vertical mixing can be observed via TKE (Fig. 8d). Positive signals are restricted to the low levels during the day (09, 12 and 15 UTC), with the latter time showing indications for increased mixing reaching midlevels. All hours from 18 UTC to 09 UTC show decreased TKE below 600 hPa and hardly any change at all above that. One needs to bear in mind, however, that the mixing through convection is not reflected in TKE fields in PARAM. Nevertheless, the PARAM signals, at least at low levels, are in clear contrast to EXPL (Fig. 7d) where TKE increases everywhere. These differences are strong indicators that the interplay between PBL turbulence, shallow and deep convection fundamentally differs between the two model configurations. Particularly during nighttime, PARAM shows a slight stabilization in the temperature profile (Fig. 8a) above 925 hPa that appears to suppress turbulence generation in this layer. This cooling may be related to the enhanced NLLJ (Fig. 8f), but it is not clear why this effect does not work in EXPL, where an even more enhanced NLLJ and also a stabilization is observed (Figs. 7a and f). The change in mixing have profound impacts on many low-level fields, whereas more agreement between EXPL and PARAM is found at mid- and upper-levels, except for some changes in the diurnal cycle.

> Overall this discussion demonstrates the enormous importance of vertical transport and mixing in a moist tropical environment where the PBL, low clouds and deep convection are closely coupled through radiative effects.
* * *
> p.21, L3: You didn't state the sign of the modification of low clouds, but then say that it leads to substantial increases in precipitation. I'd suggest to reformulate the sentence as follows: "[..] how moderate reductions in low-cloud opacity [...]".

Done as suggested.
* * *
> p.22, L7: I wouldn't use the word "impressive" here, especially as you stress in other parts of the manuscript that a quantitative interpretation of the results is questionable. Maybe just use "an increase of 560%". The same is true for p.30, L13 ("an impressive factor of 5!"). I also don't really like the use of the word 'enormous' (e.g. p.25, L24; or p.31, L26), but that might be a matter of taste.

We deleted the adjective impressive in "in the northern half of the DACCIWA box corresponds to an impressive 560%, while the southern half... " but kept it in the sentence "Particularly in the northern half of the modification region, rainfall increases by an impressive factor of 5!" because it should stress the strong effect that this small modification causes. We replaced the first enormous by a more neutral "large" but kept the second "enormous" because we think it to be appropriate in the given context.
* * *
> p.22, L31: I don't see how the sentence "This may explain the general tendency..." fits in the discussion of the EXPL simulation here, as I assume that this might be different between explicit and parameterized convection.

In climate models, we found for this region in Hannak et al. (2017) that a too strong developed nocturnal low level jet is connected to a too low cloud cover. The discussion in the respective paragraph in our paper may give hints on the processes and explain why this happens, which could be used as an explanation for climate models, too. But you are right, since the convection is parametrized, other factors can be involved, too. Therefore, we modified the paragraph as such:

> Assuming a similar behaviour also in models with parametrized convection, these processes may explain, why an underestimation of low clouds is often found together with an overestimation of NLLJ for many climate models (Knippertz et al., 2011; Hannak et al., 2017), but this needs further study.
* * *
> p.27, Figure 13: Change legend in panel (e) to f_op=1.0 & f_op=0.1.

Thank you for spotting this!
* * *
> p.28, L 13-14: I don't understand what you mean with "effectively removing tropospheric surplus and depositing...", maybe something is missing here?

We added "surplus moisture".
* * *
*p.30, L21: I would assume that air advected from the ocean is moist, not dry. Am I missing something here?*

Due to the cold temperatures and subsidence over the ocean at this time of the year, the advected air from the ocean is in fact drier than air that has already resided over land (Schuster et al., 2013)! We added "dry subsided air from the ocean" to make clearer how this is possible.
* * *
*S1, p.1, L27-29 and Figure S1: maybe add a measure of spread between the different runs to indicate the variability.*

Done.
* * *
* * *
*Typographic suggestions:*
*p.2, L7: and ITD shift –> and the ITD shift*
Done.
*p.2, L14: Eltahier –> Eltahir*
Done.
*p.3, L4: Omit either realistically or correctly.*
Done.
*p.5, L23: allows –> allow*
Corrected.
*p.5, L24: terrain following –> terrain-following*
Done.
*p.6, L21: remove grid ("a grid spacing of 13.2 km grid...")*
Done.
*p.7, L3: first (Sect. 3.1) –> first section (Sect. 3.1)*
Done.
*p.7, L17-L18: add a "the" in front of "adjacent ... highlands"*
Corrected.
*p.9, L1: "by on the order of" –> by about*
Corrected.
*p.12, L18: from –> of*
Done.
*p.13, L21: following –> followed*
Corrected.
*p.14, L4: clod –> cloud*
Done.
*p.16, L5: by on the order –> by about*
Done.
*p.16, L8: with values –> with absolute increases*
Corrected.
*p.16, L18: remove "than"*
Done.

p.16, L18: results –> result

Corrected.

p.19, L18: hardly any change at all above –> hardly any change above

Done.

p.22, L26: with values –> with decreases

Done.

p.25, L32: aerosol-radiation or –cloud interaction –> aerosol-radiation or aerosol-cloud interaction

Corrected.

p.28, L21: impacts on higher and –> impacts on higher levels and

Done.

**Reply to referee #1**

We would like to express our gratitude for the insightful review of our study. The discussion is very helpful and gives insightful remarks and valuable recommendations. We took them into account while preparing the revised version of the manuscript.

Below, the actual comments are given in blue color.
The text added to the revised version of the manuscript is marked in green.
* * *
Major Comments:
While the explanation of how the clouds were altered in the model was very clear and innovative, I have several comments about the general setup of the model and how the simulations were performed and analyzed.
First, why was the ICON model used, and more specifically, why was a global model used for this experimental setup? For such short runs, I would think a regional or at least a nested global model would be sufficient. You state that computational cost limited your runs.

The ICON model has now become a standard tool in atmospheric science in Germany following a seamless concept (one model frame for LES, NWP and climate modelling). It is widely used both by the national weather service and by many university researchers. Forecasts from ICON have been evaluated with DACCIWA campaign observations (separate paper in preparation). As a recently developed model, it fulfills highest standards and runs very efficiently. We agree that a regional model would have been sufficient here but the very flexible nesting capability of ICON allowed us to avoid any undesirable effects at the model boundaries. A classical limited area version of ICON that could be driven with existing ICON global data is currently in preparation but did not exist when we started the study. Therefore we tested the global ICON model with a nest and found that it performed well in our test cases. We also like to stress that computational cost was not the main limiting factor for the analysis, but the large amount of data, even though we restricted the output to West Africa.
We included a short explanation on p6l11:

Currently ICON is only configured as a global model but its high flexibility in terms of one- and two-way nesting allows a regional focus without any undesirable boundary effects sometimes observed for traditional limited-area models. It  performed well compared to ERA Interim (ERA-I hereafter) in several test cases we ran for DACCIWA. The comaprison can be found in the supplementary material. The simulation period was not so     much limited by computational cost but by the large amount of output, since many different state variables had to be saved for the analysis.
* * *
Do you used fixed SSTs or an interactive ocean (or ocean surface)? Since the sim- ulations are so short, I don't expect the treatment of the ocean surface to have much effect, but it would be good to know.

SSTs are contained in the input conditions, but they are not updated during run-time. We also did not expect a strong effect because of the short simulation times. We added to the sentence on p31l12:

Note that in the NWP simulations SSTs stay largely constant during the short run time; they are initialised with ERA-I, but not updated during a 5-day simulation.
* * *
For the EXPL runs, what is the domain of the nested grid? This is probably unimportant, but could have implications as odd things can happen on the boundary between nested grids.

We tested this and asked DWD for advice (to be on the safe side) and chose the grid large enough. The innermost grid was circular (this fitted best to the ICON base grid, which covers the earth in triangles). It had a radius of 30 degree and was centered on 0°E and 13°N. This way, it covered large parts of North-Western Africa. We added on p7l7:

The nest has a circular domain centered on 0° E and 13° N with a radius of 30° such that it is large enough to avoid undesirable effects near the nest's boundary.
* * *
Finally, what period is the data analysis averaged over? Is it the final four days of the five day simulations? This would be congruent with five day simulations having a single day overlap when the model was started every fourth day, however, it should be clearly stated.

Yes, that's true. We added on p8l4:

The averages were created from the final four days of the five day simulations.
* * *
Is this period the same for analysis of both the local and regional response? The timescale of the local and regional response is an important factor in interpreting your results, so you should clearly state your averaging period for analysis and support your choice with evidence from your simulations (as in the supplementary material) and/or the literature.

The averaging period is the same for the local and the regional response. We did not change any setting in between in order to keep the two responses comparable.
We added on p22l7:

To answer these questions, we expanded the analysis of the sensitivity experiments and included the Sahel zone up to about 25° N.
* * *
When describing the ICON model base state and comparing it to obs and reanalyses it may be helpful to also know how the control simulations compare with the obs/reanalyses/Hannak et al. In terms of cloud fraction and LWP/IWP. Is it a model that produces a reasonable amount of low level clouds in the base state? The discussion on pages 8 and 9 somewhat address this, but since the paper is on clouds, it might be nice to just state how the control simulations cloud fields compare to obs/reanalyses.

We looked at that and compared the profiles of low-level cloud cover, LWP, and wind speed to ERA-I. We found the base state to be satisfactory in our study region and wrote a paragraph about this. However, because the article became too long, we decided to remove it again. Now we have put a figure into the supplementary material and describe it as follows:

In addition to the general characterization of the meteorological conditions in southern West Africa for the variables precipitation and radiation, a brief discussion about the diurnal cycle of the vertical atmospheric structure is given for the wet monsoon season July, August and September 2006.
Comparison of ICON CLIM with ERA-I and observations confirms the applicability of the ICON model for the sensitivity experiments of the main article.

Figure S1 shows average profiles of cloud cover (CLC), relative humidity (RH), horizontal wind speed v_horiz as well as specific cloud water content q_c for 00, 06, 12 and 18 UTC (corresponding to local time in our study region). At 00 UTC the NLLJ is already well established and the low-level cloud deck is beginning to form (Fig. S1a).
ICON shows a considerably stronger jet than ERA-I reaching 7 m s$^{-1}$ at 925-950 hPa and consistently lower values in CLC, RH and q_c. In contrast to ERA-I, ICON tends to concentrate cloud water in the upper parts of the cloud deck around 850 hPa.
The relatively small differences in RH between the two datasets (more than 90\% from 830 hPa downwards) in contrast to differences in CLC and q_c illustrates a substantial sensitivity to the subgrid-scale cloud scheme or possibly differences in spatial variance of RH,
as the dependance of CLC on RH is quadratic in ICON. The tendency of stronger NLLJ and less cloud was also found in many climate models (Hannak et al. 2017). At midlevels around 560 hPa ICON shows a secondary peak in v_horiz, CLC, RH and q_c not found in the overall smoother and moister ERA-I profiles.

At 06 UTC the NLLJ is very similar to 00 UTC but the low-level cloud deck increases markedly in cover and q_c accompanied by an increase in RH to values well above 95 % below 900 hPa (Fig. S1b). Maximum CLC occurs at 950 hPa reaching 25% in ICON and about 45% in ERA-I, which is more realistic (cf. van der Linden et al., 2015).
Overall the discrepancies between the two models are qualitatively similar to 00 UTC (Fig. S1a). At midday (Fig. S1c), radiative heating lifts and dissolves the low-level cloud deck shifting the maximum in CLC and RH to 850 hPa, where a pronounced peak in q_c develops. Surface heating and turbulent mixing markedly slows down the low-level jet (e.g. 4.5 m s$^{-1}$ in ICON) and decreases RH to under 90% below 900 hPa with ICON being substantially drier and less cloudy in that layer.
Finally at 18 UTC (Fig. S1d) the low-level jet starts re-accelerating, keeping the generally higher values in ICON found at all times of day. The deep daytime mixing has reduced CLC and q_c and created an almost vertically constant offset between the two modeling systems. RH is already increasing at this time of day, particularly in ICON, where also the sharp gradient in v_horiz suggests a beginning decoupling of the surface. Such an early evening  transition is consistent with observations as documented in Fig. 3d in Schuster et al. (2013).
The comparison between the two datasets shows considerable biases at all times of day with generally higher low- and midlevel wind maxima in ICON but moister and more cloudy low levels in ERA-I. Investigating the reasons for these discrepancies is beyond the scope of this paper but the overall agreement in vertical structure and diurnal cycle suggests that sensitivities tested with ICON should be qualitatively meaningful.
* * *
The entire paragraph at the end of Page 12 and beginning of Page 13 is a bit confusing and unclear. (Page 12, Line 26 through Page 13, Line 5) Why do you connect the discrepancy in the longwave component with a dynamical response, but not the discrepancy in SSI? Could the change in upper-level clouds account for this difference?
It is important to point out that July 2006 might not have been the most average year, but I'm a bit perplexed by why that can explain SSI and not OLR. Maybe I'm missing

something here.

The main point was to find out, how much of the change in cloudiness could be attributed to the dynamical response in the fully non-linear model as compared to the radiative effect of the static cloud cover itself. Therefore we compared with the study by Hill et al. that only takes the latter into account. Unfortunately it is not easy to compare, because the time periods considered are not the same. This is why the characteristics of July 2006 relative to the climatological context matter. In addition, of course, clouds simulated by ICON can deviate from what CERES sees in reality. Therefore we did not make a statement about SSI because it is not possible to disentangle the two problems. For longwave is easier to interpret because there are qualitative changes (the changing sign of the variation in OLR).

We reworded the section on p13l25 to read:

> The most plausible explanation is that the relatively dry July 2006 had overall less mid- and high-level clouds than the June–September 2006–2010 average used in Hill et al., leading to a largely consistent but relatively larger effect of modifying low-level cloudiness (consistent with Fig. 9 in Hill et al., 2018). This makes it hard to distinguish the purely radiative signal from the fully nonlinear dynamical response of the atmosphere. The latter is more distinguishable in the longwave component. The increase in deep convection with optically thinner low clouds in ICON PARAM leads to a decrease in OLR in the model on the order of 10 W m−2, while the radiative transfer calculations by Hill et al. show even a small increase. In contrast, for SLI the purely radiative
> effect is a marked decrease, but ICON-PARAM shows almost constant SLI, likely due to combined dynamical effects of the increase in low-level temperature, deep convective clouds and column moisture (see Fig. 12).
* * *
> Page 19, Line 15: "most striking" is a bit subjective, don't you think? I agree that it is very striking, but I was immediately more intrigued by the low level qv and qc, whose signals can be interpreted as due to changes in vertical mixing, some of which can be explained by TKE. I think describing and explaining the signals in this manner might be more causal, but this, I suppose, is more personal opinion. This is also a massive paragraph and could probably be broken up between these two features.

Reviewer 2 said exactly the same, so we rephrased it and tried to restructure the paragraph to be more readable.

> Figure 8 shows the corresponding profiles for PARAM. Despite the overall consistent signal in rainfall and radiation as documented in Fig. 4, there are many substantial differences between the two sets of experiments.
>
> Despite a larger SSI (see Fig. 4a), PARAM has a lower daytime increase in near-surface temperature, particularly at 15 and 18 UTC, suggesting a possible impact of the earlier triggering of convection in PARAM (see Fig. 5). Near surface q_v (Fig. 8b) is strongly decreased at 09 UTC, probably due to the earlier onset of PBL mixing with transparent clouds, and then strongly increased at 12 and 15 UTC, possibly due to the lack of deep mixing as in EXPL, leading to very large differences between the two sets of experiments. Combined, the changes in temperature and moisture lead to overall less pronounced changes in RH at low levels (both negative near the

surface and positive above; Fig. 8c), associated with mostly negative changes in q_c (Fig. 8e) except for 09 UTC. These explain the somewhat unexpected results for q_c discussed in the context of Figs. 7 and 6. In contrast to EXPL, PARAM operates a positive feedback mechanism, where a reduction in low cloud leads to a further reduction. This may clarify, why so many climate models show very large negative biases in cloud cover (Hannak et al., 2017).

Increased vertical mixing can be observed via TKE (Fig. 8d). Positive signals are restricted to the low levels during the day (09, 12 and 15 UTC), with the latter time showing indications for increased mixing reaching midlevels. All hours from 18 UTC to 09 UTC show decreased TKE below 600 hPa and hardly any change at all above that. One needs to bear in mind, however, that the mixing through convection is not reflected in TKE fields in PARAM. Nevertheless, the PARAM signals, at least at low levels, are in clear contrast to EXPL (Fig. 7d) where TKE increases everywhere. These differences are strong   indicators that the interplay between PBL turbulence, shallow and deep convection   fundamentally differs between the two model configurations. Particularly during nighttime, PARAM shows a slight stabilization in the temperature profile (Fig. 8a) above 925 hPa that appears to suppress turbulence generation in this layer. This cooling may be related to the enhanced NLLJ (Fig. 8f), but it is not clear why this effect does not work in EXPL, where an even more enhanced NLLJ and also a stabilization is observed (Figs. 7a and f). The change in mixing have profound impacts on many low-level fields, whereas more agreement between EXPL and PARAM is found at mid- and upper-levels, except for some changes in the diurnal cycle.

Overall this discussion demonstrates the enormous importance of vertical transport and mixing in a moist tropical environment where the PBL, low clouds and deep convection are closely coupled through radiative effects.
* * *
The first panel in Figure 9 is really fantastic! However, I feel that the aspect of it that discusses the differences between ICON and TRMM as well as the description in the text (Page 21, Line 10 through Page 22, Line 1 "As already discussed . . . gap between the rainfall maxima") should be moved to Section 3.1 where you are discussing the differences in the control simulations and the TRMM. When inserted here it subtracts from the main point of the section which is to show how the perturbation effect precipitation into the Sahel.

Thank you. We agree that it distracts from the main topic and inserted the averages into figure 2, where the description was already given. We added the following lines:

The small middle panels in Fig. 2 show the rainfall dependent on latitude but averaged over 8° W-8° E as further analysed in section\ \ref{precip3}. The rainfall maxima over the Niger Delta region and along the coast of Guinea, Sierra Leone and Liberia are not captured in this average, which explains the rather small values between 7-10° N.

We kept the first panel in Figure 9 because it is important for the reader to see the absolute decline in rainfall when traveling northwards.
* * *
Page 22, Line 11: "This may suggest that modulations to the WAM allow a slightly deeper penetration of rainfall into the continent but one month is probably too short to make any definite statements on this area." I am a bit confused by this statement. You

are suggesting that only looking at July 2006 might not be enough to make a robust statement?

We agree that this is a bit misleading and reworded the passage to now read:

However, given that in the northern Sahel rainfall is usually caused by few distinct, intense convective systems and that soil moisture perturbations become increasingly important, five-day simulations during one month is probably insufficient to make any definite statements for this area.
* * *
Page 25, Lines 32-34: "An interesting implication . . . no significant regional impacts." This statement makes it sound like you are generalizing your result to other regions, how do you support that claim? Or are you focusing on the response to just altering the clouds in the DACCIWA region? Your statement in the summary (beginning Page 31, Line 10 with "Therefore these results . . .") is much more well supported.

Here we refer only to the DACCIWA region, we made this clearer and the lines read now (on p27l1):

An interesting implication of this result is that whatever change in aerosol-radiation or aerosol-cloud interaction is caused through changes in anthropogenic emissions in the DACCIWA region, it will likely have measurable local impacts but probably no significant ramifications elsewhere.
* * *
Page 28, Lines 18-22: This paragraph feels out of place. Why are you concerned with how quickly the atmosphere returns to its normal state? The statement "impacts on higher and more remote regions can last days" is misleading as the supplemental material suggests that the residual impacts on remote regions are complicated by the chaotic nature of the atmosphere. I think the plot in the supplemental material is quite interesting and helps explore the timescales of the response, as well as the persistence of changes when the altered clouds are removed. It might be useful to plot the envelop of the anomalies in Figure S1, in order to show the variability between the ensemble members. This would help clarify what is a consistent response to the forcing and what is due to internal variability.

The original idea was to find out, if the experiment that we conducted has a long-lasting effect on the structure of the atmosphere, or if it is a short-lived phenomenon. In the first case, an influence on remote regions would appear more likely. We agree that the chaotic nature of the system makes it difficult to discuss the life-time effect. We plotted the respective figure again and included the envelope in the bias. Since the envelope stays quite constant with time after the switch-off for the boundary layer variables, we believe that our statements are correct. We modified the passage on p29l24 in the main manuscript in the following way:

Impacts on higher levels and more remote regions can last for days but the signals hardly stand out from the high level of background variations indicating the chaotic nature of the atmosphere.

And in the supplementary material in the figure caption of Fig. S1:

And added to the corresponding text on p4l13 Where we had already referred to the chaotic nature of the system:

The latter is reflected in the growing shaded areas denoting the envelope of all runs in the bias in the right-hand side panels of Fig. \ref{longr}.
* * *
What is the remote (Sahel) response of precipitation for the positive values of f_op = 3 and 10 in the PARAM experiments? I realize these are extreme values, but it would be interesting to see if the same signals appear.

In the Sahel there is not too much difference noticeable due to the already low absolute values of precipitation. The DACCIWA-box, on the other hand, dries out further as already indicated in Fig. 4c. Below please find a figure (middle panel  control run, right panels  f_op = 3, left panel f_op=0.1) that illustrates the regional distribution of this drying.

[Figure]

and the same plot as figure 9 in the paper, but with f_op=3 and f_op=10 included:

[Figure]
* * *
Minor Comments & Typographic Corrections:
Many passages would be improved by the introduction of commas, especially the Oxford comma. I've identified a few below, but if you'd like, I would be glad to mark up the submitted manuscript with where I believe commas would help separate clauses and help clarify the manuscript.

Thank you, we tried our best to improve the punctuation.
* * *
Page 1, Line 7: effect

Done.

Page 1, Line 24: "do not show skillful forecasts of precipitation for the next days", is awkward, perhaps "do not produce skillful short term precipitation forecasts" is clearer.

Agreed.

Page 2, Line 9: "determined by the WAM system", is unclear, what about the WAM system determines these characteristics? "are connected within the WAM system" emphasizes the challenge of the interconnectedness of the WAM.

That's better, thank you.

Page 2, Line 32: I would remove "which is currently gaining increasing attention".

Done.

Page 3, Line 17: misrepresenting

Done.

Page 3, Lines 19-30: I would reorganize this entire paragraph to flow better.
This study is part of the Dynamics-Aerosol-Chemistry-Cloud Interactions in West Africa (DACCIWA) project (Knippertz et al., 2015) that aims to better understand the consequences of the rapid increase of anthropogenic emissions in West Africa on the local air quality, weather and climate. To the best of our knowledge, it is the first to analyze the radiative impact of the low-level cloudiness over southern West Africa on the thermodynamics and dynamics of the regional atmospheric system in a fully non-linear and systematic way. The analysis is based on a number of targeted sensitivity experiments using the numerical weather prediction model ICON (Icosahedral Nonhydrostatic), systematically changing the optical thickness of the model clouds. This allows us to clarify the impact of the inter-model spread in cloudiness found in Hannak et al. (2017) on the overall monsoon development in both parameterized and explicit regimes of convection. Although aerosols are not directly modeled in our experiments, the effects found for imposed changes of cloud optical thickness also help to understand variations of the natural system brought about by aerosol effects on cloud properties and radiation, which in a similar way control the amount of shortwave radiation reaching the surface or interact with clouds through modifications in the diurnal cycle of the PBL (e.g. Deetz et al., 2018a).

Thank you so much, this is really appreciated and we put it into the manuscript.

Page 6, Line 4: Green's

Done.

Page 6, Line 7: (Dee et al., 2011), and do not use data assimilation. (the comma helps separate the idea that the simulations are not using assimilation instead of ERA-I,

Done.

Thank you.

Figure 2: maybe outlining the box in each subplot would help highlight the similarities
and differences
Since reviewer #1 had an issue with this figure, it was modified and now contains a box in each
subplot.

Done.

Figure 4: why is the axis dimension for SLI so large? All the other subplots are so well
framed, but this one has so much empty space it looks weird. Maybe 400-430 W/mˆ2
would suffice?
That was actually done on purpose. The range is similar to the other radiative quantities  to illustrate
that the variability is not so large for SLI. Therefore, we would like to keep it this way.

Done.

Done.

Done.

Corrected.

Figure 10: The legend for subplot (e) should have fop = 0.1 for the red curve.
We totally overlooked that, thank you!

Done.

We added on p27l3:

This quite noticeable impact was found in the simulations with explicitly simulated convection. The influence of parametrization of convection in this experiment will be discussed next.

Page 28, Line 9: remove "just mentioned"

Done.

Section 4, First Paragraph: The second sentence introduces the idea of representation of clouds in models, but then immediately returns to the idea of describing the clouds in reality. The paragraph might flow better this way:
In the present study, we analyzed the role of low-level clouds over southern West Africa on the local meteorology and larger monsoon system. They frequently form during the night close to the surface and often persist long into the following day. At their maximum diurnal extent, they cover a vast area of about 850 000 km2 in southern West Africa (van der Linden et al., 2015). Their formation is linked to cold advection and turbulent mixing associated with the NLLJ and radiative cooling (Schrage and Fink, 2012; Schuster et al., 2013; Kalthoff et al., 2018). These clouds play an important role in the energy budget and diurnal cycle during summertime and tend to be badly represented in many climate models (Hannak et al., 2017). The role of these clouds in the WAM system was assessed here for the first time in a fully nonlinear way via sensitivity experiments using the ICON model from the DWD in NWP mode for July 2006.

We would like thank you very much for this rephrasing, we changed the paragraph accordingly.

Supplemental Material, Page 1, Line 4: Would read better as: "To investigate this we use EXPL experiments, in which f_op = 0.1 is applied for the first 4 days . . ." The phrase "as in EXPL" makes it unclear what type of experiment is being done here.

Thank you, we adapted it.

[revised manuscript text omitted]

**S1    Supplementary Material**

**S1.1    Climatology and model evaluation**

In addition to the general characterization of the meteorological conditions in southern West Africa for the variables precipitation and radiation, a brief discussion about the diurnal cycle of the vertical atmospheric structure is given for the wet monsoon

season July, August and September 2006. Comparison of ICON CLIM with ERA-I and observations confirms the applicability of the ICON model for the sensitivity experiments of the main article.

Figure S1 shows average profiles of cloud cover ($CLC$), relative humidity ($RH$), horizontal wind speed $v_{horiz}$ as well as specific cloud water content $q_c$ for 00, 06, 12 and 18 UTC (corresponding to local time in our study region). At 00 UTC the NLLJ is already well established and the low-level cloud deck is beginning to form (Fig. S1a). ICON shows a considerably stronger jet than ERA-I reaching 7 ms$^{-1}$ at 925-950 hPa and consistently lower values in $CLC$, $RH$ and $q_c$. In contrast to ERA-I, ICON tends to concentrate cloud water in the upper parts of the cloud deck around 850 $hPa$. The relatively small differences in $RH$ between the two datasets (more than 90% from 830 $hPa$ downwards) in contrast to differences in $CLC$ and $q_c$ illustrates a substantial sensitivity to the subgrid-scale cloud scheme or possibly differences in spatial variance of $RH$, as the dependance of $CLC$ on $RH$ is quadratic in ICON. The tendency of stronger NLLJ and less cloud was also found in many climate models (Hannak et al. 2017). At midlevels around 560 hPa ICON shows a secondary peak in $v_{horiz}$, $CLC$, $RH$ and $q_c$ not found in the overall smoother and moister ERA-I profiles.

At 06 UTC the NLLJ is very similar to 00 UTC but the low-level cloud deck increases markedly in cover and $q_c$ accompanied by an increase in $RH$ to values well above 95 % below 900 hPa (Fig. S1b). Maximum $CLC$ occurs at 950 hPa reaching 25% in ICON and about 45% in ERA-I, which is more realistic (cf. van der Linden et al., 2015). Overall the discrepancies between the two models are qualitatively similar to 00 UTC (Fig. S1a). At midday (Fig. S1c), radiative heating lifts and dissolves the low-level cloud deck shifting the maximum in $CLC$ and $RH$ to 850 hPa, where a pronounced peak in $q_c$ develops. Surface heating and turbulent mixing markedly slows down the low-level jet (e.g. 4.5 ms$^{-1}$ in ICON) and decreases $RH$ to under 90% below 900 hPa with ICON being substantially drier and less cloudy in that layer. Finally at 18 UTC (Fig. S1d) the low-level jet starts re-accelerating, keeping the generally higher values in ICON found at all times of day. The deep daytime mixing has reduced $CLC$ and $q_c$ and created an almost vertically constant offset between the two modeling systems. $RH$ is already increasing at this time of day, particularly in ICON, where also the sharp gradient in $v_{horiz}$ suggests a beginning decoupling of the surface. Such an early evening transition is consistent with observations as documented in Fig. 3d in Schuster et al. (2013). The comparison between the two datasets shows considerable biases at all times of day with generally higher low- and midlevel wind maxima in ICON but moister and more cloudy low levels in ERA-I. Investigating the reasons for these discrepancies is beyond the scope of this paper but the overall agreement in vertical structure and diurnal cycle suggests that sensitivities tested with ICON should be qualitatively meaningful.

**S1.2 Temporal stability of opacitiy-induced effects**

An additional aspect to be discussed is the response time of the atmospheric system to the imposed cloud modifications. To investigate this we use EXPL experiments, in which $f_{op} = 0.1$ is applied for the first 4 days but then switched off for 6 more days of simulation time. Control runs with $f_{op} = 1.0$ for all times were produced for comparison. As in EXPL, simulations were started every 4th day but run out to 10 days and two starting dates in August were added (3rd and 7th of August) to give better statistics for the time evolution. Figure S2 shows box-averaged 10-day time-series of $RR$, cover of low clouds $CLC_{low}$ and that of high clouds $CLC_{high}$ (below 800 hPa and above 400 hPa, respectively) for the DACCIWA region (Figs. S2a–c). The corresponding differences between the two sets of simulations are provided in the right-hand side panels of Fig. S2.

After the switch-off at 12 UTC on the fifth simulation day (i.e. after 96 hours), the differences in $SSI$ and low-level $T$ are reduced almost immediately (not shown), but for other variables the response is slower. $RR$ shows the enhancement of afternoon and evening precipitation for $f_{op} = 0.1$ as in EXPL (Figs. S2a and f). The enhancement is still fully visible for the first 14 hours after switch-off, indicating that the influence of the forcing during the morning hours is already enough to generate more instability and trigger more convection later in the day. After that, differences between the two runs become negligible. Before the switch in $f_{op}$, $CLC_{low}$ shows the familiar afternoon decrease and nighttime increase (Figs. S2b and g). On the day of the change, some signal remains until the morning of the following day, similar to $RR$. The small but on average slightly positive differences after that may be a reflection of increased surface fluxes after the strongly enhanced rainfall of the first five days. $CLC_{high}$ (Figs. S2c and h) shows a considerably slower response. Differences between the two runs need one full diurnal cycle to establish and are then positive for the next three days. After the switch, there is a marked decrease

[Figure]

**Figure S2.** Time-series of $RR$ (a), $CLC_{low}$ (b) and $CLC_{high}$ (c) of the experiments with $f_{op} = 1.0$ all the time (blue line) and $f_{op}$ switched from 0.1 to 1.0 after 96 hours (orange line). Results are averaged over the DACCIWA box, for all 10 runs in July-August 2006. (d) and (e) depict similar averages of $RR$ but for the northern boxes 15–20°N, 8°W–8°E and 15–20°N, 8°W–8°E, respectively. The corresponding differences are shown as green lines in (f)–(j), where the shaded area denotes the maximum and minimum values of the time-series. Switch-off time is indicated by a grey line in all panels.

in differences but then an overall tendency for relatively large values for two more days. This is consistent with Raymond et al. (2011), who show a considerably longer response time in the tropical upper troposphere than at low levels for a given perturbation.

Another interesting question is the impact on regions to the north, i.e. downstream of the DACCIWA box with respect to the monsoon flow. Figures S2d, e, i and j show corresponding plots for $RR$ over the Sahelian regions 10–15°N and 15–20°N, both averaged over 8°W–8°E. For the former, again an initial response time of about one day is observed followed by a period of small positive differences. During the last five days of the simulation there is then no clear net difference between the two sets of experiments but much larger fluctuations. As these do not follow a strict diurnal cycle, we speculate that this is mostly a reflection of the overall chaotic nature of the atmosphere growing with leadtime. This conclusion is consistent with the similar behavior found for the 15–20°N band.

So in summary, this experiment shows that low-level variables such as $SSI$ and $T$ react almost immediately to changes in low cloud during the day. Low-level cloud cover and rainfall respond after one full diurnal cycle, while upper-level variables and neighboring regions show even longer responses, but also increasingly chaotic behavior. The latter is reflected in the growing shaded areas denoting the envelope of all runs in the bias in the right-hand side panels of Fig. S2.

**S1.3   Table of acronyms**

**Table S1.** Table of acronyms and abbreviations used in the main article.

| Acronym | denotation |
|---|---|
| $\theta_e$ | equivalent potential temperature |
| CALIPSO | Cloud-Aerosol Lidar and Infrared Pathfinder Satellite Observation |
| CERES | Clouds and the Earth's Radiant Energy System |
| CM SAF | Satellite Application Facility on Climate Monitoring |
| $CLC$ | cloud cover |
| $CLC_{low}$ | low-cloud cover |
| $CLC_{high}$ | high-cloud cover |
| CloudSat | satellite in A-train with cloud profiling radar |
| COSMO model | Consortium for Small-scale Modeling model |
| COSMO-EU | regional COSMO model for Europe |
| DACCIWA | Dynamics-Aerosol-Chemistry-Cloud Interactions in West Africa project |
| DWD | German weather service |
| EBAF-Surface | Energy Balanced And Filled surface irradiance |
| EBAF-TOA | Energy Balanced And Filled top of atmosphere irradiance |
| ECMWF | European Centre for Medium-Range Weather Forecasts |
| ERA-I | ERA Interim |
| EXPL | experiment with explicit convection |
| $f_{op}$ | opacity factor |
| GERB | Geostationary Earth Radiation Budget |
| GPCP | Global Precipitation Climatology Project |
| GPCC | Global Precipitation Climatology Centre |
| ICON | Icosahedral Non-hydrostatic ( numerical weather prediction model of DWD) |
| IFS | Integrated Forecasting System |
| ITD | Intertropical Discontinuity |
| MODIS | Moderate Resolution Imaging Spectroradiometer |
| MPI-M | Max Planck Institute for Meteorology |
| MVIRI | Meteosat Visible and Infrared Imager |
| NLLJ | nocturnal low-level jet |
| NPP | Suomi National Polar-orbiting Partnership |
| NWP | numerical weather prediction |
| $OLR$ | outgoing longwave radiation |

| Acronym | denotation |
|---|---|
| $OSR$ | outgoing shortwave radiation |
| $p_{sfc}$ | surface pressure |
| PARAM | experiment with parameterised convection |
| PBL | planetary boundary layer |
| $q_c$ | cloud liquid water content |
| $q_i$ | cloud ice content |
| $q_v$ | specific humidity |
| $RH$ | relative humidity |
| $RR$ | precipitation rate |
| RRTM | Rapid Radiation Transfer Model |
| SARAH | Surface Solar Radiation Data Set Heliosat |
| SEVIRI | Spinning Enhanced Visible and InfraRed Imager |
| SLEVE | smooth level vertical (coordinate) |
| $SLI$ | surface longwave irradiance |
| SOCRATES | Suite Of Community RAdiative Transfer codes based on Edwards and Slingo |
| $SSI$ | solar surface irradiance |
| $SST$ | sea surface temperature |
| SWA | southern West Africa |
| SWITCH | experiment where opacity factor is switched to 1 after some time |
| $T$ | temperature |
| TERRA | soil and vegetation model |
| $TKE$ | turbulent kinetic energy |
| TMPA | TRMM Multisatellite Precipitation Analysis |
| TOA | top of atmosphere |
| TRMM | Tropical Rainfall Measuring Mission |
| $v_{horiz}$ | horizontal wind speed |
| WAM | West African monsoon |